# Two Effects, One Trigger: On the Modality Gap, Object Bias, and Information Imbalance in Contrastive Vision-Language Models

**Simon Schrodi**[*,1]   **David T. Hoffmann**[*,1,2]   **Max Argus**[1]   **Volker Fischer**[2]   **Thomas Brox**[1]
[1]University of Freiburg, [2]Bosch Center for Artificial Intelligence

## Abstract

Contrastive vision-language models (VLMs), like CLIP, have gained popularity for their versatile applicability to various downstream tasks. Despite their successes in some tasks, like zero-shot object recognition, they perform surprisingly poor on other tasks, like attribute recognition. Previous work has attributed these challenges to the modality gap, a separation of image and text in the shared representation space, and to a bias towards objects over other factors, such as attributes. In this analysis paper, we investigate both phenomena thoroughly. We evaluated off-the-shelf VLMs and while the gap's influence on performance is typically overshadowed by other factors, we find indications that closing the gap indeed leads to improvements. Moreover, we find that, contrary to intuition, only few embedding dimensions drive the gap and that the embedding spaces are differently organized. To allow for a clean study of object bias, we introduce a definition and a corresponding measure of it. Equipped with this tool, we find that object bias does not lead to worse performance on other concepts, such as attributes per se. However, why do both phenomena, modality gap and object bias, emerge in the first place? To answer this fundamental question and uncover some of the inner workings of contrastive VLMs, we conducted experiments that allowed us to control the amount of shared information between the modalities. These experiments revealed that the driving factor behind both the modality gap and the object bias, is an information imbalance between images and captions, and unveiled an intriguing connection between the modality gap and entropy of the logits.

## 1 Introduction

Contrastive Vision-Language Models (VLMs) are successfully applied to numerous tasks. They benefit from their ability to exploit weak supervision in contrastive pre-training (Radford et al., 2021; Jia et al., 2021), which can be acquired by scraping image-text pairs from the internet. In spite of this, they exhibit intriguing properties: strong zero-shot image recognition performance (Radford et al., 2021; Menon & Vondrick, 2023), cross-modal understanding (Chen et al., 2023a), retrieval (Ma et al., 2022), or robustness (Nguyen et al., 2022). Despite such remarkable advancements, our understanding of the representations learned by VLMs is still limited. Liang et al. (2022) identified a modality gap in the shared embedding space and Bravo et al. (2023) conjectured about a bias towards objects. However, how do these phenomena affect downstream performance, why do they emerge, and can we mitigate them? To answer these questions and enhance our understanding of the learned representations, we thoroughly study both, modality gap and object bias.

The **modality gap** is a geometric phenomenon characterized by the two modalities lying in completely separate regions of the shared embedding space of contrastive VLMs. Liang et al. (2022) attributed its emergence to the cone effect during model initialization with the contrastive loss preserving the gap. Subsequent work studied the influence of the temperature parameter (Udandarao, 2022; Shi et al., 2023). Intuitively, one would expect the gap to limit performance, but despite previous efforts, the impact of the modality gap on performance has remained unclear so far: Is the gap even a relevant problem worth fighting? Similarly, the interaction between closing the gap post-hoc

---

[*]Equal contribution. Correspondence to: {schrodi, hoffmann}@cs.uni-freiburg.de. Code is available at
https://github.com/lmb-freiburg/two-effects-one-trigger.

and its effect on performance also remained elusive. Beyond these performance-related questions, we aim to more thoroughly understand the modality gap: Are all dimensions contributing equally? Are the modalities structured similarly, in a sense that neighborhood relations are similar? Do the representations of both modalities have similar meaning? Lastly, we investigate whether the modality gap is rather a bug or a feature. Despite the importance of these questions, they have not been thoroughly addressed so far. Our analysis offers answers to these questions and equips practitioners with actionable insights to address various issues related to the modality gap.

Beyond the modality gap, recent work (Bravo et al., 2023; Trager et al., 2023) found significantly worse performance of contrastive VLMs for attribute tasks compared to object tasks, which led them to the hypothesis that they are **biased towards objects** (Bravo et al., 2023). However, "bias towards objects" was previously not formally defined and was only assessed by the poorer performance. For a thorough study of object bias and its emergence, we go beyond the previous notion of bias towards objects and introduce a novel metric to measure object bias: Matching Object Attribute Distance (MOAD). It assesses the bias towards objects compared to other factors, such as attributes. Beyond assessing bias towards objects or attributes, MOAD's generic formulation allows to also study other types of biases in learned representations. Equipped with this new metric, we can now answer questions about object bias: Does object bias even affect performance on non-object (attribute) tasks? Are objects just more often mentioned in captions, leading to the object bias? Answering these questions, can guide us to effective strategies to overcome such biases of VLMs.

Finally, we investigate the fundamental question of what actually leads to the emergence of the modality gap and object bias. We identify a common cause: **information imbalance**. Information imbalance refers to the availability of more information in one modality than the other. For example, captions are often sparse (lossy), focusing on the most salient object(s), while images hold far more information not captured by their captions; see Figure 1. Since the caption is unknown to the image encoder, it cannot know what information of the image it needs to encode to align the image encoding with the text encoding. The best that the image encoder can do is to focus on the most salient parts of the image, i.e., the parts that are typically present in the captions. As a result, the image encoder de-

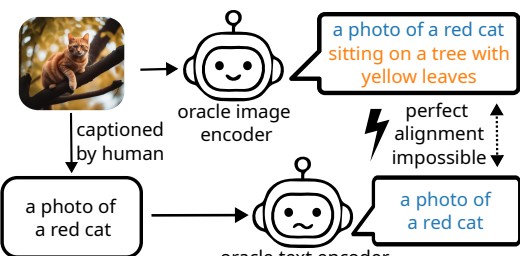

Figure 1: **Illustration of information imbalance** between images (top left) and captions (bottom left). This imbalance makes it even for an oracle image encoder virtually impossible to predict the content of a caption, leading to undesirable effects in contrastive training, such as the modality gap and object bias (see Section 6).

velops a bias towards these parts. For natural language captions, these are typically object names. Besides that, the modality gap also emerges as a by-product of contrastive training under an information imbalanced regime. Here, the model trades off alignment—which is limited due to the information imbalance—with uniformity by making all images and all texts more dissimilar. We thoroughly validate our hypothesis by manipulating information imbalance in experiments on synthetic as well as real data. Our insights equip researchers with the understanding needed to craft, for example, data filtering or caption enrichment strategies to reduce the modality gap or object bias.

The findings of our **analysis** paper are: 1) For off-the-shelf contrastive VLMs, a larger modality gap correlates with better performance due to common confounders. However, controlling for these confounders suggests that a *lower modality gap correlates with better performance*. 2) *Only few embedding dimensions* drive the modality gap. 3) Image and text embeddings have *distinct characteristics*, like different neighborhood orderings. 4) Object bias does not negatively correlate with performance on object tasks. 5) An *information imbalance* between the modalities leads to both the modality gap and the object bias. 6) The modality gap is a by-product of the model's efforts to *deal with the uncertainty* (entropy) caused by information imbalance. 7) Object bias is caused by higher *per-sample caption presence bias*, which is also a consequence of the information imbalance.

## 2 RELATED WORK

Contrastive VLMs have emerged as an effective approach to learn representations through weak supervision that work for a wide range of tasks and have intriguing properties, such as strong zero-

shot abilities. However, the learning framework comes with potential issues and our understanding of it is in its infancy.

For example, recent work showed the existence of a *modality gap*. They attributed it to the cone effect at model initialization and the contrastive loss that preserves it (Liang et al., 2022). The cone effect refers to the observation that the embedding space is restricted to a narrow cone. The modality gap emerges at initialization, as it is unlikely that both encoders use the same cone with random initialization. Subsequent work studied the influence of the temperature parameter (Udandarao, 2022; Shi et al., 2023). Other work found that the modality gap is orthogonal to the span of image and text embeddings (Zhang et al., 2023). In this work, we find that few dimensions drive the modalities apart and show that information imbalance is the main factor for the emergence of the modality gap.

Geirhos et al. (2021); Li et al. (2023b) found that large contrastive VLMs close the gap to human perception, but others found several failure modes of them (Yuksekgonul et al., 2022; Brody, 2023). The importance of data was studied by Nguyen et al. (2022); Xu et al. (2024), generalization/robustness by Mayilvahanan et al. (2024); Crabbé et al. (2023), the learned features were analyzed by Goh et al. (2021); Materzyńska et al. (2022); Rashtchian et al. (2023), compositionality was studied by Jia et al. (2021); Couairon et al. (2022); Trager et al. (2023), and learned abilities and (social) biases by Agarwal et al. (2021); Yamada et al. (2023); Zhang et al. (2022); Wu & Maji (2022); Shtedritski et al. (2023); Hamidieh et al. (2023). Lastly, Bravo et al. (2023) hypothesized that VLMs may be *biased towards objects*. Here, we validate that VLMs are biased towards objects but find that VLMs with smaller object bias may not necessarily perform better on attribute tasks.

## 3  EXPERIMENTAL SETUP

In this work, we performed a series of experiments using both real as well as fully-controllable, synthetic data. Below, we briefly outline the common experimental details. We describe the specific experimental settings in the respective sections.

**Pre-trained contrastive VLMs.** We used a total of 98 contrastive VLMs such as CLIP (Radford et al., 2021) or SigLIP (Zhai et al., 2023) for our analysis. The VLMs were pre-trained on various datasets. Refer to Appendix A for the full list. We distinguished between VLMs pre-trained on medium- (i.e., dataset size of $\leq 140\,\mathrm{M}$) and large-scale datasets. For in-depth analysis, we used the pre-trained CLIP ViT-B/16 (Radford et al., 2021) and SigLIP ViT-B/16 (Zhai et al., 2023).

**Evaluation protocols.** We evaluated the pre-trained contrastive VLMs on MS COCO (Lin et al., 2014; Chen et al., 2015), ImageNet (Russakovsky et al., 2015), MIT-States (Isola et al., 2015), and UT-Zappos (Yu & Grauman, 2014) with the standard evaluation protocols from the literature. We prepended the prompt `"a photo of"` to the description, object class, or attribute, following Radford et al. (2021). For evaluation, we used R@1 for zero-shot image-to-text retrieval or top-1 accuracy for zero-shot object and attribute recognition. Further details on the datasets and evaluation protocols are provided in Appendix A.

**Fully-controlled, synthetic data.** To study the effect of information imbalance on the modality gap and object bias, we built a fully-controllable dataset based on Morpho-MNIST (Castro et al., 2019), called Multi-modal Attributes and Digits (MAD). We used the following morphing or warping operations as latent factors (i.e., attributes) from Castro et al. (2019): thickness, swelling, fractures. We further added scaling, colors, and captions. Figure 2 and Appendix B provide examples. To generate captions, we mapped the digit class and the other factors to words and chained them together in random order, e.g., `1-thickening-swelling-fractures-large-blue`. We embedded

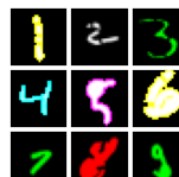

Figure 2: **Examples from MAD**.

each factor to a single token. To study the effect of information imbalance, we varied the number of attributes within each caption while, importantly, keeping the images unchanged. For example, if we restrict each caption to one attribute (in addition to the always present object), above (full) caption becomes, e.g., `1-blue` or `1-large`. Model and training details are provided in Appendix B.

**Experiments on real data.** To validate our hypothesis in a realistic setting, we trained CLIP RN50 models on the image-text dataset CC12M (Changpinyo et al., 2021), and manipulated the captions. We provide further details in Section 6 and Appendix F.1.

Figure 3: **Relation between modality gap (L2M & RMG, larger value → larger gap) and down-stream performance** for a total of 98 contrastive VLMs pre-trained on medium- and large-scale datasets (each scatter point is a VLM). The plots indicate no to weak positive correlations between performance and modality gap (see the numbers in Table 1).

Table 1: **Kendall's $\tau$ rank correlation between downstream performance and various factors** for models trained on medium and large datasets. ✓denotes statistical significance ($p < 0.05$). Model, embedding, and dataset size correlate stronger with performance than the modality gap.

| Downstream task | Modality gap (L2M) | Modality gap (RMG) | Model size | Embedding size | Dataset size |
|---|---|---|---|---|---|
| MS COCO | 0.167 (✗) / 0.148 (✗) | 0.083 (✗) / -0.007 (✗) | -0.354 (✗) / 0.579 (✓) | 0.264 (✗) / 0.62 (✓) | -0.129 (✗) / 0.252 (✓) |
| ImageNet | -0.008 (✗) / 0.28 (✓) | -0.109 (✗) / 0.169 (✓) | -0.5 (✓) / 0.62 (✓) | 0.318 (✗) / 0.668 (✓) | -0.034 (✗) / 0.206 (✓) |

## 4 PARTING WITH FALSE INTUITIONS ABOUT THE MODALITY GAP

Liang et al. (2022) showed the existence of a *modality gap*; a geometric phenomenon of the shared embedding space of multi-modal models, such as contrastive VLMs. Specifically, they showed that the embeddings of each modality lie in *completely separate* regions. They proposed that the gap exists at model initialization (cone effect) and that the contrastive loss preserves it. They proposed the L2-distance between the Means (L2M) to measure the gap: $\text{L2M} := ||\frac{1}{N}\sum_{i=1}^{N}\mathbf{x}_i - \frac{1}{N}\sum_{i=1}^{N}\mathbf{y}_i||$, where $\mathbf{x}_i$ is the $i$-th L2-normalized image embedding and $\mathbf{y}_i$ the $i$-th text embedding.

Despite Liang et al.'s (2022) and subsequent work (Udandarao, 2022; Shi et al., 2023; Zhang et al., 2023), many questions about the modality gap remain unanswered so far: What is its impact on downstream performance? How does it manifest itself in the embeddings? We will answer these questions in the following: We discuss the relationship between the modality gap and downstream performance (Section 4.1) and find that only few embedding dimensions drive it (Section 4.2). In contrast to the cone hypothesis (Liang et al., 2022), the gap emerges even when we initialize with no gap and find evidence that the contrastive loss can narrow the gap (see Appendices F.4 and F.5). Thus, the cone effect can't be the main factor. What else leads to the emergence of the gap? We show in Section 6 that an information imbalance is the main factor that leads to its emergence.

### 4.1 DOES A SMALLER MODALITY GAP LEAD TO BETTER PERFORMANCE?

The effect of the modality gap on downstream performance has been discussed controversially (So et al., 2023; Zhou et al., 2023; Liang et al., 2022; Zhang et al., 2023). We intuitively expect that a smaller modality gap leads to better performance. However, do we observe this also in practice?

To answer this question, we compared the downstream performance and the modality gap across 98 contrastive VLMs, e.g., CLIP or SigLIP, pre-trained on various datasets such as LAION or WebLI. We evaluated the downstream performance on MS COCO for image-to-text retrieval (text-to-image retrieval yielded similar results) as well as on ImageNet zero-shot image classification. Using L2M to measure the gap and particulary to compare the gap across models comes with some limitations. For instance, L2M neglects whether an image-text pair matches, and it does not take the effectively used space into account; refer to Appendix D.1 for a detailed discussion. To address these short-comings of L2M, we propose the Relative Modality Gap (RMG) measure:

$$\text{RMG} := \frac{\frac{1}{N}\sum_{i=1}^{N}d(\mathbf{x}_i, \mathbf{y}_i)}{\frac{1}{2N(N-1)}\left(\sum_{i,j=1;i\neq j}^{N}d(\mathbf{x}_i, \mathbf{x}_j) + \sum_{i,j=1;i\neq j}^{N}d(\mathbf{y}_i, \mathbf{y}_j)\right) + \frac{1}{N}\sum_{i=1}^{N}d(\mathbf{x}_i, \mathbf{y}_i)} \quad , \quad (1)$$

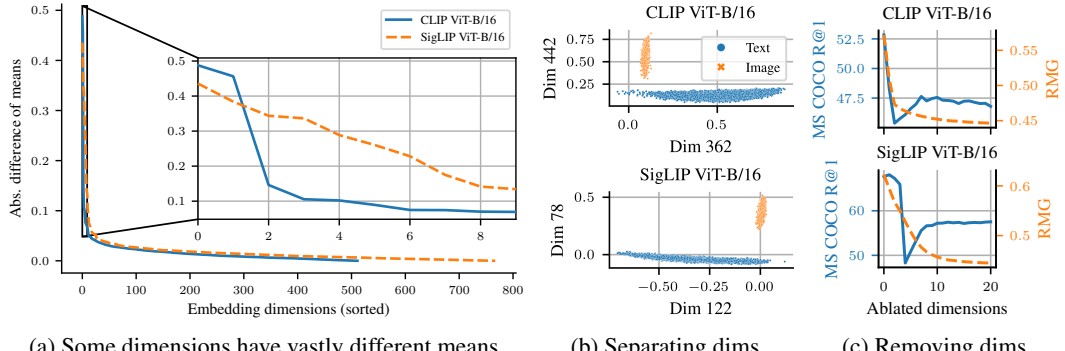

(a) Some dimensions have vastly different means.     (b) Separating dims.     (c) Removing dims.

Figure 4: **Few embedding dimensions separate the modalities.** Results on MS-COCO. (a) We plot the absolute difference in the means of each embedding dimension between the modalities. Most dimensions have similar means for both modalities, but for some the differences are huge. (b) Pairs of these high difference dimensions can perfectly separate the modalities (we show the ones with largest mean for each modality). (c) Successive removal of embedding dimensions based on the sorting of embedding dimensions from (a) leads to a sharp drop, followed by a partial recovery of downstream performance, while the modality gap gradually closes (similar results for L2M). See Appendix D.3 for results on ImageNet and the plots in (b) with the largest two dimensions of (a).

where $\mathbf{x}_i, \mathbf{y}_i$ are the $i$-th L2-normalized image or text embeddings, respectively, and $d$ is some distance function (we used cosine dissimilarity scaled to [0,1]). Intuitively, the numerator measures the gap where it matters, i.e., for matching image-text pairs, and the denominator accounts for the effectively used space through an approximation using the intra-modality distances. We added the distances of matching image-text pairs to the denominator to scale RMG to $[0, 1]$.

Figure 3 and Table 1 show that a larger modality gap counter-intuitively correlates with better downstream performance. However, does this imply that a larger modality gap *leads to* better performance? No, as Table 1 reveals, other factors such as model or embedding size have a stronger effect on performance. It appears that the negative impact of the modality gap is overshadowed by these other factors. Indeed, when we control for these factors, we find weak indications for the expected negative correlation for some of the pre-training datasets, i.e., a smaller gap seems to correlate with better performance (see Appendix D.2). Thus, the modality gap is a problem worth fighting.

> **Takeaway 1:** A larger modality gap has mild positive correlation with downstream performance. However, there is no indication that a larger modality gap leads to a better performance; rather, it suggests the presence of common confounders (e.g., model size).

## 4.2 Few embedding dimensions drive the modality gap

To obtain better insights into the nature of the modality gap, we studied two questions: 1) Is the modality gap present in all dimensions or only a subset thereof, and 2) does post-hoc closing of the modality gap improve downstream performance?

**Do all embedding dimension contribute to the gap?** To study the influence of each embedding dimension, we compared their means. Figure 4a shows that while most embedding dimensions have very similar means, interestingly, some have stark differences. When we plot the dimensions with the largest mean in each modality, we find that they suffice to perfectly separate the modalities (Figure 4b). Further, they have large variance within one modality but only negligible variance in the other. In summary, Figures 4a and 4c show that these dimensions are the main contributors.

> **Takeaway 2:** Few embedding dimensions drive the modality gap. We find that two dimensions suffice to perfectly separate the modalities.

**Can we close the gap post-hoc?** One could suspect that removing above embedding dimensions will close the modality gap and as a result yield better performance. To test this, we successively

Table 2: **Dissimilarity of neighborhood orderings in the embedding space** using normalized Kendall-$\tau$ distance $\in [0, 1]$. Higher normalized Kendall-$\tau$ distances indicate that the ranking of neighbors differs more. We used three different ImageNet-100 splits.

| | CIFAR-10 | CIFAR-100 | ImgNet-100-1 | ImgNet-100-2 | ImgNet-100-3 |
|---|---|---|---|---|---|
| CLIP ViT-B/16 | 0.3399 | 0.4965 | 0.4975 | 0.5046 | 0.5081 |
| SigLIP ViT-B/16 | 0.5044 | 0.4981 | 0.5003 | 0.4965 | 0.4987 |

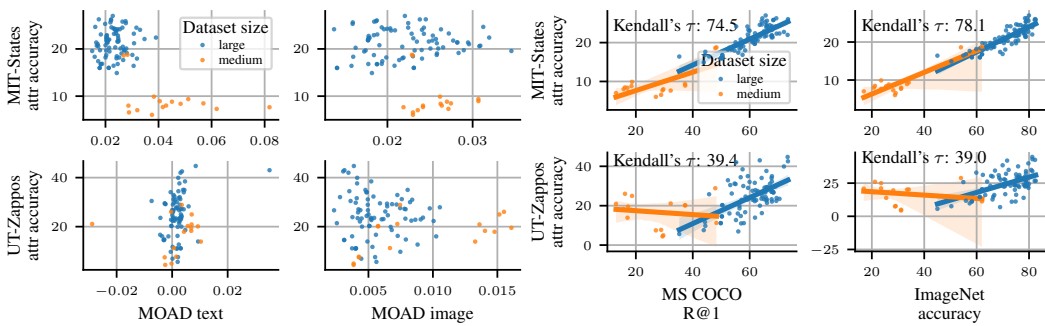

(a) Object bias vs. downstream performance.     (b) Object vs. attribute performance.

Figure 5: **Object bias and performance on attribute tasks.** (a) We find a bias towards objects (positive MOAD values) but no correlation with attribute performance. We attribute this to the (b) positive correlation between performance improvements on object tasks and attribute tasks.

ablated embedding dimensions (i.e., set their values to zero) based on the sorting from Figure 4a, re-normalized, and evaluated their downstream performance. Figure 4c shows that the modality gap and the downstream performance sharply decreases, followed by a *partial* recovery of the performance. The behavior on downstream performance is explained by the substantial change in cross-modal neighborhoods caused by the ablation and re-normalization of the largest and thereby most influential embedding dimensions on the geometry of the embedding space. Refer to Appendix D.4 for an illustrative example. Thus, while removing the largest contributors closes the gap, it leads to degraded performance.

Similar to the above, other post-hoc approaches tested in previous work also did not lead to consistent improvements: Liang et al. (2022) show that computing the "modality gap vector" and closing the gap by adding this vector generally hurts performance. Similarly, for the ideal words approach of Trager et al. (2023) we find that simple shifting closes the gap, increases average similarities, but does not improve performance (refer to Appendix D.5 for details). But why can we not just close the gap post-hoc with such translation approaches to improve the performance?

A key precondition of these translation approaches are similar neighborhood relations across modalities. To test their similarity, we computed the mean embedding of each class in CIFAR-10, CIFAR-100 (Krizhevsky et al., 2009), and three ImageNet-100 splits (Hoffmann et al., 2022). We computed the Kendall-$\tau$ distance normalized by the number of classes. Intuitively, it counts the percentage of bubble-sort swaps w.r.t. all possible swaps needed to transform modality A's nearest neighbor list to match modality B's. Table 2 shows that the neighborhood orderings are dissimilar between the modalities. Hence, these simple transformations cannot improve performance.

> **Takeaway 3:** Simple post-hoc approaches can close the modality gap but do not improve performance. One reason for this is that the modalities have different local neighborhoods.

Looking at the bigger picture, why does the modality gap emerge in the first place? In Section 6, we will show that an information imbalance between modalities is the main factor for its emergence.

## 5 OBJECT BIAS IS A CAPTION PRESENCE BIAS

The hypothesis that contrastive VLMs are biased towards objects was based on their poorer performance on non-object (e.g., attribute) compared to object tasks (Bravo et al., 2023). However, this can be misleading, as an attribute-based task could just be more difficult than an object-based task.

Thus, to study the bias towards objects, we propose a measure for object vs. attribute bias, denoted as Matching Object Attribute Distance (MOAD). MOAD quantifies how well a model can distinguish matching from non-matching images (or texts) of objects $o$ compared to attributes $a$. Matching images (texts) show the same object or attribute, whereas non-matching images (texts) show different objects or attributes. MOAD can be computed for each modality. For images, we define it as:

$$\text{MOAD}_{\text{img}} := \frac{1}{2|O|} \sum_{\text{obj} \in O} \left( \text{sim}_{\text{obj}}^+ - \text{sim}_{\text{obj}}^- \right) - \frac{1}{2|A|} \sum_{\text{att} \in A} \left( \text{sim}_{\text{att}}^+ - \text{sim}_{\text{att}}^- \right), \quad (2)$$

where $O$ denotes the object classes, $\text{sim}_{\text{obj}}^+$ denotes the mean similarity between the L2-normalized image embeddings that entail the same object obj, $\text{sim}_{\text{obj}}^-$ the mean similarity between different objects (i.e., the image does not entail the same object) and $\text{sim}_{\text{att}}^+$ & $\text{sim}_{\text{att}}^-$ are similarly defined for attributes $A$. Positive values of MOAD indicate a bias towards objects, negative values a bias towards attributes, and zero no bias. Refer to Appendix E.1 for the detailed definition. Intuitively, MOAD measures whether the visible/present objects or attributes have a larger influence on the similarity of two images/captions. Note that MOAD is not specific to objects and attributes, we provide a generic formulation, called Bias measure via Relative Angle Change of Embeddings (BRACE) , in Appendix E.2.

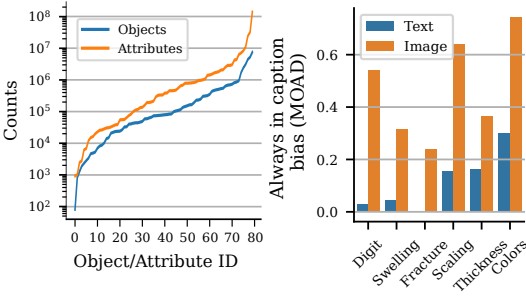

(a) Object and attribute counts in LAION-2B.
(b) We can bias the model to any factor of choice.

Figure 6: **Object bias is caused by a per-sample caption presence bias.** (a) Object bias is not caused by the word frequency as attributes appear more frequently than objects. (b) We trained CLIP models on MAD and changed which factor is always in the caption (i.e., caption presence bias). The models are biased towards whichever factor (e.g., color, thickness, etc.) that we made prevalent in the captions.

Similar to the analysis on the modality gap in Section 4, we first investigated the relation between object bias and attribute performance. Figure 5a shows that the majority of contrastive VLMs exhibit a bias towards objects (positive MOAD values), as expected. Notably, models trained on large-scale data are less biased towards objects (smaller positive values) compared to models trained on medium-scale data. However, we find no clear correlation between the bias towards objects and performance; especially for models trained on large-scale data.

Thus, reducing the object bias does not appear to affect attribute performance—but what does? Figure 5b shows medium-to-strong correlations between performance on object and attribute tasks. Thus, it seems that simply improving VLMs (without specific focus on attributes) also improves their performance on attributes as a by-product.

> **Takeaway 4:** Contrastive VLMs trained on large-scale data tend to have a lower object bias than models trained on medium-scale. However, there is no clear relation between object bias and attribute performance. This can be attributed to the observation that performance improvements on object tasks correlate with improvements on attribute tasks.

**Is object bias explained by the global word frequencies of the dataset?** One may suspect that the word frequency of the training dataset (i.e., prior distribution over words $p(\text{word})$) causes the bias towards objects. However, Figure 6a disproves this, as attributes are actually mentioned more often than objects in the LAION-2B's captions (Schuhmann et al., 2022) (we used object and attribute classes from Bravo et al. (2023)). Thus, we hypothesize that the bias towards objects arises from the per-sample prevalence of objects in natural language, i.e., humans tend to describe the most salient

object(s) and typically only few of their attributes in a caption. In other words, we propose that the conditional probability of words given an image $p(\text{word}|\text{image})$ is what leads to the bias.

To verify this, we used MAD (Section 3) and redefined the prevalence of words in the caption. Specifically, we trained five contrastive VLMs for which we varied the prevalent factor by making it always present in the caption and sampling one of the other factors randomly by chance. Figure 6b validates that each model is biased towards whichever factor was prevalent during its training (larger positive MOAD values). We also find that the bias is larger for the image encoder. This is expected, as it needs to match to the most likely caption, while the text encoder can simply encode the entire information (that information is likely always present in the image), as sketched in Figure 1 and outlined in detail in Section 6. We provide further experimental evidence in Appendix E.3.

> **Takeaway 5:** Bias towards concepts, e.g., objects, is caused by their high probability of appearing in captions (given that said concept is present in an image), rather than by their overall frequency in the dataset.

Naturally, the question that arises is how we can reduce the object bias. We will address this in the next section.

## 6 INFORMATION IMBALANCE TRIGGERS MODALITY GAP AND OBJECT BIAS

The previous sections analyzed both the modality gap and bias towards objects. However, what causes their emergence in contrastive VLMs? In this section, we will show that an *information imbalance* in the data is the main cause of both. Information imbalance refers to unequal amount of information transmitted by the modalities, i.e., while an image contains a lot of information, its caption is typically incomplete; as illustrated in Figure 1. Specifically, it mostly only entails one or two of the most salient objects and a handful of other factors such as attributes of these objects.

### 6.1 HOW DOES INFORMATION IMBALANCE CAUSE THE MODALITY GAP AND OBJECT BIAS?

**The origin of the object bias.** A consequence of the information imbalance in the data is that the encoders can hardly align their embeddings, as they have no way of knowing what information is available in the other modality. To achieve sufficient alignment[1] (numerators in the CLIP loss (Equation 4)) despite the uncertainty from the information imbalance, the best both encoders can do (especially the image encoder) is to focus on the factors that are most likely present in the caption (or image)—typically objects—while using a smaller scale or fewer dimensions for factors that are less likely to be present (e.g., attributes). By that, the average cosine similarity will be less affected when the unlikely factors are not present. This leads to the caption presence bias discussed in Section 5, i.e., a bias towards the most likely present words given the images.

**The origin of the modality gap.** We also hypothesize that the main factor for the emergence of the modality gap is an information imbalance between modalities in the data. This imbalance makes it difficult for the encoders to align their embeddings, as discussed above. As a result, maximizing alignment for matching image-text pairs (numerators of the CLIP loss (Equation 4)) becomes challenging. In other words, the alignment term is bounded. With alignment being bounded (and optimized), the only way to further reduce the total loss is by maximizing uniformity (the denominators). Consequently, contrastive VLMs tend to focus more on increasing the distance of non-matching pairs, i.e., maximizing the uniformity. After all, the contrastive loss maximizes the similarity of matching image-text pairs *relative to* non-matching pairs. In other words, VLMs compromise on alignment, which is inherently limited by the information imbalance, to achieve higher uniformity. Since uniformity operates only across modalities, as in the CLIP loss, VLMs push the modalities apart, ultimately leading to the gap. We conjecture it does so by using few dimensions (e.g., the ones in Figures 4a and 4b) which substantially increase uniformity but have a small effect on alignment.

**Experimental validation in a fully-controlled synthetic setting.** To validate that information imbalance triggers both the modality gap and the object bias, we manipulated information imbalance

---

[1]For our explanation, we use the alignment and uniformity terms of the contrastive loss (e.g., CLIP loss Equation 4) defined by Wang & Isola (2020). Refer to Appendix C for background.

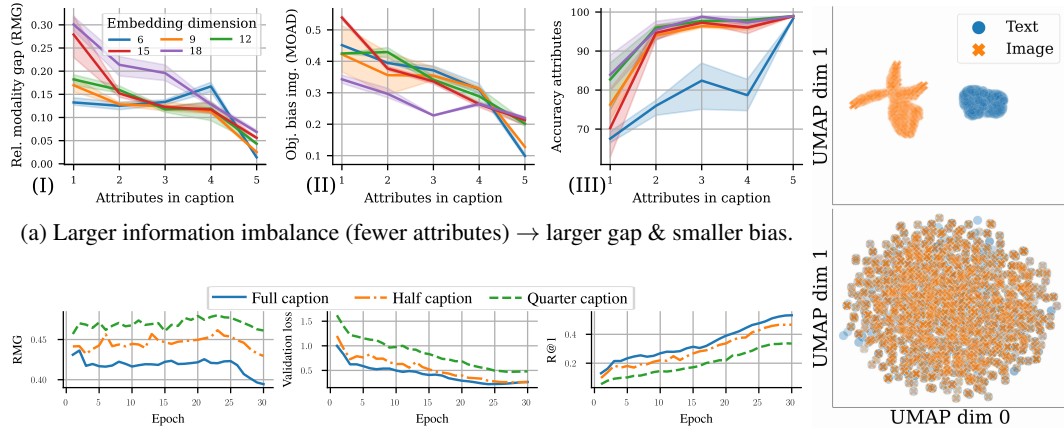

(a) Larger information imbalance (fewer attributes) → larger gap & smaller bias.

(c) Larger information imbalance (shorter captions) → larger gap (real data).          (b) UMAP embeddings.

Figure 7: **Increasing shared information between modalities (smaller information imbalance) improves the representations.** (a) To study the influence of information imbalance between the modalities, we manipulated the number of attributes in the captions in MAD (the images are always affected by all attributes). As the amount of shared information between the modalities increases, the modality gap (I) and bias towards objects reduce (II), while downstream accuracy improves (III). We show additional results in Figure 13. (b) The contrastive loss is able to close the modality gap given full shared information (no information imbalance) between modalities, as illustrated by the UMAP embeddings (McInnes et al., 2018) at model initialization (top) and after training (bottom). (c) To verify our explanation also on real data, we trained CLIP models on CC12M and created an information imbalance by dropping ½ or ¾ of the captions. Similarly, we find that lower information imbalance leads to a smaller modality gap. Additional results are provided in Appendix F.1.

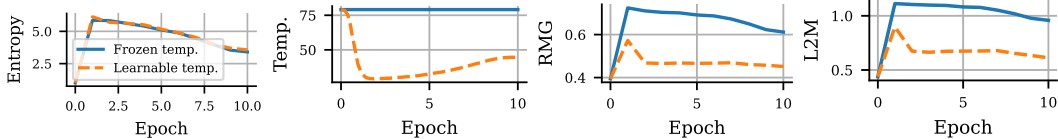

Figure 8: **A model trained with frozen temperature increases the modality gap more than a model with trainable temperature to achieve a similar logit entropy.**

in a fully-controlled synthetic experimental setting. Specifically, we varied the number of attributes included in the captions of MAD, while the object (digit) was always included. Importantly, we only manipulated the captions and left the images unchanged. The information imbalance is larger with fewer attributes and smaller with more attributes. Section 3 provides further experimental details.

Figure 7a shows that both the modality gap and object bias decrease as we reduce information imbalance. This validates our hypothesis. Notably, even when the modality gap is large at model initialization, contrastive VLMs can close it in the full information setting (Figure 7a(I) and qualitatively in Figure 7b). This is aligned with our hypothesis. In contrast, the cone effect hypothesis is not consistent with our observation. Further, we find that the image encoders are more biased towards objects than the text encoders (Figure 5a, 6b and 13(III-IV)). This is expected, as the text encoders know what to encode, whereas the image encoders must model the uncertainty over the captions' content. Finally, we find that performance improves as information imbalance decreases (Figure 7a(III)), and that our findings are consistent across different embedding dimensions.

**Experimental validation on real data.** The main advantage of the synthetic setting above is that we have full control over the interventions and avoid confounding factors. To verify that our findings also apply to real data, we trained three CLIP models on CC12M, either with the "full" captions or omissions of ½ ("half") or ¾ ("quarter") contiguous parts of them; see Appendix F.1 for details. The omissions increase the information imbalance. It is important to note that this can also be viewed in reverse: the "full" captions represent enriched versions (i.e., less information imbalance) of the "half" or "quarter" captions, similar to work on caption enrichment (Fan et al., 2023), but

without introducing hallucinations. Figure 7c confirms that the modality gap is smaller with lower information imbalance. Thus, our hypothesis (Section 6.1) also holds on real data.

> **Takeaway 6:** An information imbalance between modalities leads to both modality gap and object bias. Reducing this imbalance decreases both the modality gap and object bias.

## 6.2 IS THE MODALITY GAP A BUG OR A FEATURE?

Figure 7c shows that the validation losses for "full" and "half" captions are very similar at the end of training, but retrieval performance (R@1) is clearly higher for "full" captions. The lower performance for "half" captions suggests that the model does not align positives (w.r.t. their negatives) as well and makes false retrievals more likely. However, why is this not reflected in the validation loss? A possible explanation is a different logit entropy, which directly influences the loss values. The entropy over the logits models the entropy (uncertainty) of the data.

To change the (global) entropy, contrastive VLMs have a learnable temperature parameter ($\tau$ in the CLIP loss (Equation 4)). However, changing the modality gap may also change the entropy. For example, reducing the similarity between a pair of positives while keeping the negatives identical (increasing the gap), increases the entropy. Thus, changes to the embeddings that increase the modality gap also increase the entropy. Consequently, the training can control the entropy in the output logits (i.e., the uncertainty) in two ways: via the temperature or the embeddings (e.g., by changing the modality gap). One way to achieve this is by changing a few embeddings, as in Section 4.2. The ability to also change the entropy via the modality gap[2] equips the model with the flexibility to change the entropy independently for each sample. In contrast, the temperature parameter only affects the entropy globally.

**Experimental validation.** To investigate whether CLIP changes the modality gap to change the entropy, we first trained CLIP on CC12M with the "full" captions (low information imbalance). We subsequently fine-tuned this model on the "quarter" captions (increased information imbalance, resulting in higher entropy) in two conditions: with learnable temperature (*learn. temp.*) or frozen (*frozen temp.*). Note that "learn. temp." can adjust both the temperature and modality gap, but "frozen temp." can only adjust the gap. Appendix F.2 provides further experimental details.

Following our explanation in Section 6.1 and the hypothesis that CLIP "increases the gap to adjust the entropy", we expect low entropy and modality gap at the start of fine-tuning followed by a sharp increase. Importantly, we would expect that entropy is similar for both, "frozen temp" and "learnable temp." As a consequence, the modality gap should be larger when the temperature is kept frozen. Indeed, we observe that "frozen temp." increases the modality gap significantly more to match the same entropy (Figure 8). Thus, the modality gap can be interpreted as a feature, not a bug, as it adds flexibility to the model to change the entropy.

> **Takeaway 7:** During training, models learn to estimate the entropy (uncertainty) of the data. For contrastive VLMs, changes to the embeddings that lead to a higher/smaller modality gap lead to a higher/lower entropy of the logits. This increases the model's flexibility in controlling logit entropy.

## 7 CONCLUSION

In this work, we showed that both, the modality gap and object bias of contrastive VLMs, are triggered by an information imbalance between modalities. This information imbalance limits the achievable alignment in the embedding space. We showed that the effect of the modality gap on performance is typically overshadowed by other factors, and it is driven by only few embedding dimensions. We find that the modality gap is a symptom of changes to the embedding that lead to higher logit entropy. Further, we confirmed that contrastive VLMs have a bias towards objects, and could relate it to a per-sample caption presence bias due to information imbalance. These insights will serve researchers and practitioners in understanding and modifying the learned representations of contrastive VLMs.

---

[2]For simplicity, we refer to this as "increasing/decreasing the gap to increase/decrease the entropy". However, we do not claim a causal relationship; the changes can also just be measurable by the modality gap.

ACKNOWLEDGMENTS

This research was funded by the German Federal Ministry for the Environment, Nature Conservation, Nuclear Safety and Consumer Protection based on a resolution of the German Bundestag (67KI2029A), the German Research Foundation (DFG) under grant numbers 417962828 and 499552394 (SFB 1597), and the Bosch Center for Artificial Intelligence.

We would like to thank Philipp Schröppel and Sudhanshu Mittal for helpful comments on the draft, Maria A. Bravo and Simon Ging for insightful discussions, and Simon Ging and Elias Kempf for preparing the CC12M and CC3M datasets.

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

# A  FURTHER DETAILS FOR THE ANALYSIS EXPERIMENTS OF PRE-TRAINED VISION-LANGUAGE MODELS

**List of contrastive VLMs.**  For our large-scale analyses, we considered a total of 112 contrastive VLMs trained across various datasets provided by OpenCLIP (Ilharco et al., 2021; Cherti et al., 2023)[3] before filtering (see below). We considered the following contrastive VLMs with the following characteristics:

- **CLIP variants**: OpenAI's CLIP (Radford et al., 2021), OpenCLIP (Cherti et al., 2023), MetaCLIP (Xu et al., 2024), CLIP-A (Li et al., 2023a), EVA-CLIP, EVA-02-CLIP (Sun et al., 2023), CoCa (Yu et al., 2022), NLLB-CLIP (Visheratin, 2023), or SigLIP (Zhai et al., 2023; Alabdulmohsin et al., 2023).

- **Vision backbones:** ResNet (He et al., 2016), ConvNeXt (Liu et al., 2022), or ViT (Dosovitskiy et al., 2020).

- **Pretraining datasets:** OpenAI's proprietary ($400\,$M) WebImageText dataset (Radford et al., 2021), LAION-$400\,$M, LAION-$2\,$B, LAION-Aesthetic ($900\,$M) (Schuhmann et al., 2022), Merged-$2\,$B (merge of $1.6\,$B samples from LAION-$2\,$B and $0.4\,$B samples from COYO-$700\,$M (Byeon et al., 2022)) (Sun et al., 2023), WebLI (Chen et al., 2023b), So-$400\,$M (Alabdulmohsin et al., 2023), MetaCLIP ($400\,$M) (Xu et al., 2024), Conceptual $12\,$M (Changpinyo et al., 2021), YFCC ($15\,$M) (Thomee et al., 2016), CommonPool-s (max. $12.8\,$M; refer to Table 3 of Gadre et al. (2023) for the details of filtering), CommonPool-m (max. $128\,$M), CommonPool-l (max. $1.28\,$B), CommonPool-xl (max. $12.8\,$B) (Gadre et al., 2023), and DataComp-s ($1.4\,$M), DataComp-m ($14\,$M), DataComp-l ($140\,$M), DataComp-xl/DataComp-1B ($1\,$B) (Gadre et al., 2023).

OpenCLIP contains a large variety of models, some of which, e.g., have poor performance or are redundant. Thus, we filtered models using the following criteria:

- We removed models that were specifically fine-tuned, e.g., on MS COCO.

- When there were models that were trained on the same dataset but saved at different checkpoints, we selected the latest checkpoint.

- We excluded models that achieve $< 5\%$ R@1 on MS COCO, under the assumption that such low scores indicate some sort of failure that is likely to distort the analyses.

After filtering, we used 98 (out of 112) contrastive VLMs for our analyses.

**Dataset details.**  We ran our evaluations on MS COCO (Lin et al., 2014; Chen et al., 2015), ImageNet (Russakovsky et al., 2015), MIT-States (Isola et al., 2015), and UT-Zappos (Yu & Grauman, 2014). The datasets comprise 25000 (5000 images with 5 captions each), 50000, 12995, or 2914 test samples, respectively. MS COCO and ImageNet are standard datasets for evaluation of retrieval or object recognition performance, respectively. We used the standard evaluation protocols to image-to-text retrieval performance (text-to-image retrieval yielded similar results) or computed top-1 accuracy. MIT-States consists of 245 objects and 115 adjectives (attributes), while UT-Zappos consists of 12 shoe types with 16 fine-grained states ($\sim$ attributes). For both datasets, we assume that we do not know the object of a respective image and only want to find the adjective or fine-grained state. We considered this a classification problem, following previous work (Trager et al., 2023). Note that these datasets implicitly assume that the adjectives are mutually exclusive per image. However, this may not be necessarily true, as multiple adjectives or fine-grained states may be present in the image.

**Evaluation prompts.**  For MS COCO, we prepended the prompt `"a photo of"` to the description of each image following Radford et al. (2021). For ImageNet, we used the prompts `"a photo of a {obj}"` (Radford et al., 2021). For both MIT-States and UT-Zappos, we used the prompts `"an image of a {attr} object"`.

---

[3] https://github.com/mlfoundations/open_clip

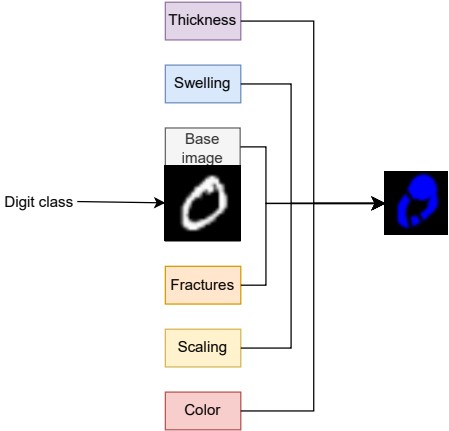

Figure 9: **Causal graph of MAD.**

## B    FURTHER DETAILS FOR THE EXPERIMENTS ON THE MULTI-MODAL ATTRIBUTES AND DIGITS DATASET

**Multi-modal Attributes and Digits (MAD).**   Our dataset Multi-modal Attributes and Digits (MAD) is based on the MNIST (LeCun, 1998) variation Morpho-MNIST (Castro et al., 2019). The causal graph of the data-generating process of MAD is depicted in Figure 9.   We used the following words for digits (0, ..., 9), altering image thickness (thickening, thinning, no thickthinning), swelling (swelling, no swelling), fractures (fracture, no fracture), scaling (large, small), and color (gray, red, green, blue, cyan, magenta, yellow).  Thus, we have 10 digits and 16 attributes. Figure 10 provides examples of image-text pairs of MAD.

In our experiments, we investigated information imbalance in the captions by restricting the number of attributes present within each caption. We provide examples below, where we sequentially remove the amount of information within the captions, i.e., fewer latent factors (attributes) are present in the caption:

- Full information setting (i.e., digit & all five attributes)
    - yellow-swelling-thickening-9-large-fracture
    - swelling-thickening-6-red-small-fracture
    - 5-large-yellow-no swelling-fracture-thinning
- Partial information setting I (i.e., digit & four attributes)
    - yellow-swelling-thickening-9-large
    - swelling-thickening-6-red-small
    - 5-large-yellow-no swelling-fracture
- Partial information setting II (i.e., digit & three attributes)
    - yellow-swelling-thickening-9
    - swelling-thickening-6-red
    - 5-large-yellow-no swelling
- Partial information setting III (i.e., digit & two attributes)
    - yellow-swelling-9
    - swelling-thickening-6
    - 5-large-yellow
- Partial information setting IV (i.e., digit & one attribute)

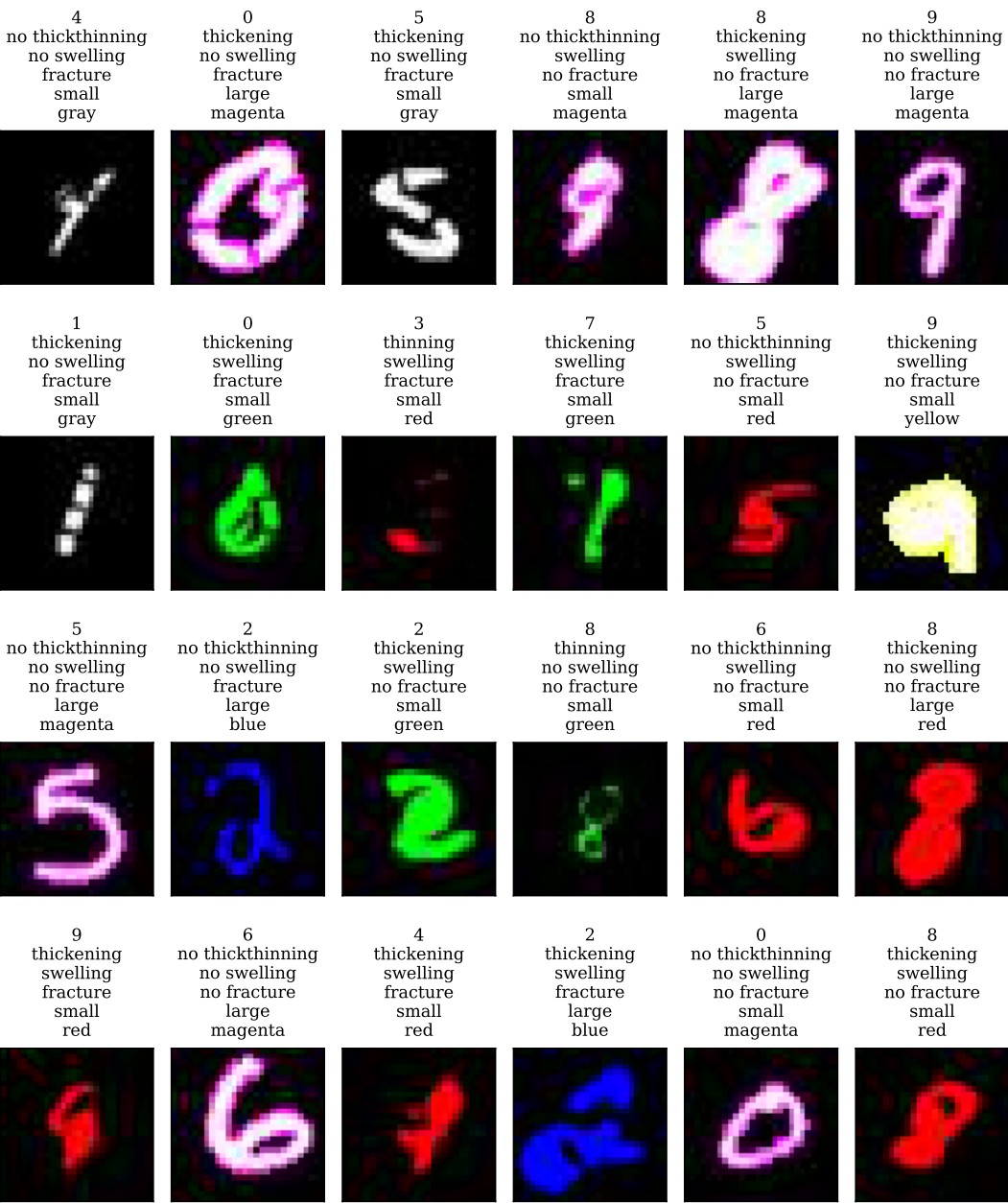

Figure 10: **Example images with corresponding caption of our MAD dataset.** For example, the image in the first row and first column shows the digit 4 without altering the thickness, no swelling applied, with fracture augmentation, scaled down and the color gray. Note that the words of the captions are shuffled (not in this figure for convenience).

- yellow-9
- swelling-6
- 5-large

Note that while all the latent factors, i.e., digit and all five attributes, still affect the generated image, the caption may only provide partial information, i.e., attributes are missing from the caption but not the image.

**Model details for MAD experiments.** We used small CLIP models. Specifically, the ViT-based vision backbone comprises 6 layers, each with a dimensionality $d$ of 256 and $\lfloor d/64 \rfloor = 4$ heads.

The transformer-based language backbone also comprises 6 layers, each with a dimensionality of 256 and 8 heads. We set the patch size to 7 and context length to 8. The vocabulary consists of 28 words, i.e., all the words for digits (10) and attributes (16), as well as a start and end symbol (2).

**Training details.** We trained all models with a batch size of 128 for 200 epochs with a learning rate warm-up period of 5 epochs. We used AdamW (Loshchilov & Hutter, 2019) as optimizer with cosine annealing learning rate schedule (Loshchilov & Hutter, 2019). We always selected the best performing learning rate across 3 learning rates $\{5 \cdot 10^{-4},\ 5 \cdot 10^{-5},\ 10^{-5}\}$ each trained with 3 random seeds. The best learning rate was selected by comparing average of the average ideal word accuracy and average zero-shot accuracy on all attributes and the class label. For all of our results, we report the average over 3 random seeds.

## C    BACKGROUND ON THE CLIP LOSS

Contrastive representation learning (Chopra et al., 2005; Gutmann & Hyvärinen, 2010; Sohn, 2016; Oord et al., 2018) leverages paired inputs as weak supervision signal. The basic idea is to learn representations in a shared representation space that are as similar as possible for "positive/matching" pairs, while as dissimilar as possible for "negative/non-matching" pairs. A popular choice for contrastive learning approaches is the InfoNCE objective (Oord et al., 2018):

$$\mathcal{L}(f_x, f_y) := -\frac{1}{N} \sum_{i=1}^{N} \log \frac{\exp(\tau f_x(\mathbf{x}_i)^T f_y(\mathbf{y}_i)))}{\sum\limits_{j=1}^{N} \exp(\tau f_x(\mathbf{x}_i)^T f_y(\mathbf{y}_j)))}\ ,\tag{3}$$

where $f_x, f_y$ are two encoders for the inputs $\{\mathbf{x}\}_{1:N}, \{\mathbf{y}\}_{1:N}$ and $\tau$ is the scalar temperature. Wang & Isola (Wang & Isola, 2020) define two components of the loss:

- **Alignment** (numerator): matching pairs should be close, i.e., aligned.
- **Uniformity** (denominator): representations should be roughly uniformly distributed on the unit hypersphere.

For multi-modal contrastive representation learning, it is popular to use a symmetric version of above objective (Radford et al., 2021):

$$\mathcal{L}_{\text{CLIP}} = -\frac{1}{2N} \sum_{i=1}^{N} \log \frac{\exp(\tau f_x(\mathbf{x}_i)^T f_y(\mathbf{y}_i)))}{\sum\limits_{j=1}^{N} \exp(\tau f_x(\mathbf{x}_i)^T f_y(\mathbf{y}_j)))} - \frac{1}{2N} \sum_{j=1}^{N} \log \frac{\exp(\tau f_x(\mathbf{x}_j)^T f_y(\mathbf{y}_j)))}{\sum\limits_{i=1}^{N} \exp(\tau f_x(\mathbf{x}_i)^T f_y(\mathbf{y}_j)))}\ .$$

$$\tag{4}$$

We can define the alignment and uniformity term similarly for the CLIP loss. It is important to observe that the repulsive forces of the CLIP loss only act across the modalities but not within them.

## D    EXTENDED DETAILS FOR SECTION 4

### D.1    FURTHER DETAILS ON RELATIVE MODALITY GAP MEASURE (RMG)

**Limitations of L2M.** L2M has been initially proposed as modality gap distance by Liang et al. (2022). However, it has several limitations that we will discuss below:

(a) L2M does not account for difference of the effectively used embedding space.

(b) L2M takes a distributional instead of a per-sample view.

(c) The L2 norm can be sensitive to outlier embedding dimensions.

(d) The L2 norm is an unintuitive distance measure when all points lie on the surface of a sphere.

Regarding (a): note that different models can use different amounts of the unit hypersphere, as the contrastive loss accounts for relative cosine similarities. Here, the similarity is always relative to

Table 3: **When controlling for the dataset choice, the modality gap seems to correlate negatively with performance (i.e., the smaller the gap the better the performance) in most cases.** We report the Kendall's $\tau$ rank correlations between RMG (top) or L2M (bottom) and the respective downstream performance. Additionally, we report the p-values of the correlations. Note, that the p-values are sometimes large due to the small number of models we are left with for each dataset (see parenthesis) and therefore results should be taken with a grain of salt.

| | RMG | | | | |
|---|---|---|---|---|---|
| Evaluation dataset | DataComp-1B (10) | LAION-400M (7) | LAION-2B (20) | OpenAI (12) | WebLI (9) |
| MS COCO | 0.244 (p=0.375) | -0.619 (p=0.069) | -0.263 (p=0.114) | -0.143 (p=0.548) | -0.611 (p=0.02) |
| ImageNet | 0.644 (p=0.011) | -0.714 (p=0.03) | -0.168 (p=0.319) | 0.302 (p=0.195) | -0.5 (p=0.073) |
| | L2M | | | | |
| MS COCO | 0.333 (p=0.221) | -0.619 (p=0.069) | -0.189 (p=0.259) | 0.238 (p=0.304) | -0.5 (p=0.073) |
| ImageNet | 0.733 (p=0.003) | -0.714 (p=0.03) | -0.063 (p=0.738) | 0.397 (p=0.077) | -0.333 (p=0.257) |

the similarity to the negative samples. Consequently, models can use a varying degree of the unit hypersphere and, thus, L2 distances can have a different meaning. For example, consider two models that have the same L2M but the first model uses the entire unit hypersphere, while the second model only uses a small fraction of it. While L2M suggests that the modality gap distance is the same, the actual gap of the first model is significantly smaller, since the average distances between the samples are larger.

Regarding (b): intuition suggests that matching image-text pairs should be close but non-matching pairs can (and should) be large. L2M considers the distance of the means of all pairs. This can lead to misleading results. An illustrative (but unlikely) example is the case of two distributions that occupy exactly the same region of the hypersphere, but are rotated by n-degree (i.e., they are very misaligned). Clearly, there exists a gap between the modalities but L2M does not indicate it.

Lastly for (c) or (d), the L2 norm can be sensitive to embedding dimensions that exhibit vast differences and are unintuitive since CLIP's embeddings are on the unit hypersphere. For instance, our discovered most modality-separating embedding dimensions qualify for this.

**Relative Modality Gap (RMG).** As a remedy to above outlined limitations, we proposed a Relative Modality Gap (RMG) measure in the main text (see Equation 1). RMG computes the distances between matching image-text pairs instead of the means to address (b). Since density estimation in high-dimensional spaces is difficult, we used the mean distances between all samples per modality as rough approximation to address (a). Finally, we used cosine similarities instead of the L2 norm to address (c) and (d).

### D.2 FURTHER ANALYSIS ON THE RELATIONSHIP OF THE MODALITY GAP AND DOWNSTREAM PERFORMANCE

**Controlling for the dataset choice.** To mitigate the impact of the dataset, we studied the relationship between the modality gap and downstream performance for each dataset individually. For this, only datasets with at least seven models were considered: DataComp-1B (10 models), LAION-400M (7), LAION-2B (20), OpenAI's CLIP dataset (12), and WebLI (9).

Table 3 shows that when controlling for dataset choice, we observe the expected negative rank correlation (i.e., smaller modality gap correlates with better downstream performance) in seven and six out of ten cases for RMG or L2M, respectively. However, we do not find statistical significance for these correlations in most cases. Note that this is most likely due to the small number of models used.

Through further inspection, we made an interesting observation for models that were pre-trained on DataComp-1B and evaluated on MS COCO: the positive correlation we found can be attributed to the differences in training of CLIP and CLIP-A models (Li et al., 2023a). CLIP-A models utilize an inverse scaling law, where larger image/text encoders are trained on fewer image/text tokens, i.e., image resizing or masking and shorter context length for the text tokens (e.g., through truncation or

Table 4: **Data filtering makes the modality gap more narrow.** We report RMG (1st, 2nd table) and L2M (3rd, 4th) on MS COCO (1st, 3rd) and ImageNet (2nd, 4th). For all filtering strategies, the modality gap narrows after filtering the unfiltered dataset.

### MS COCO, RMG

| CommonPool scale | filtering strategy | | | | | |
|---|---|---|---|---|---|---|
| | none | basic | text-based | image-based | LAION-2B-based | CLIP-based |
| small | 0.533 | 0.462 | 0.461 | 0.445 | 0.447 | 0.461 |
| medium | 0.537 | 0.499 | 0.507 | 0.494 | 0.446 | 0.487 |
| large | 0.603 | 0.577 | 0.593 | 0.587 | 0.511 | 0.544 |
| xlarge | 0.586 | n/a | n/a | n/a | 0.531 | 0.543 |

### ImageNet, RMG

| | none | basic | text-based | image-based | LAION-2B-based | CLIP-based |
|---|---|---|---|---|---|---|
| small | 0.598 | 0.502 | 0.492 | 0.461 | 0.468 | 0.485 |
| medium | 0.595 | 0.547 | 0.553 | 0.525 | 0.478 | 0.523 |
| large | 0.651 | 0.628 | 0.628 | 0.625 | 0.557 | 0.584 |
| xlarge | 0.647 | n/a | n/a | n/a | 0.593 | 0.603 |

### MS COCO, L2M

| | none | basic | text-based | image-based | LAION-2B-based | CLIP-based |
|---|---|---|---|---|---|---|
| small | 0.733 | 0.508 | 0.515 | 0.356 | 0.346 | 0.519 |
| medium | 0.773 | 0.678 | 0.702 | 0.677 | 0.482 | 0.639 |
| large | 0.898 | 0.86 | 0.892 | 0.877 | 0.742 | 0.8 |
| xlarge | 0.875 | n/a | n/a | n/a | 0.849 | 0.862 |

### ImageNet, L2M

| | none | basic | text-based | image-based | LAION-2B-based | CLIP-based |
|---|---|---|---|---|---|---|
| small | 0.805 | 0.525 | 0.516 | 0.3 | 0.3 | 0.491 |
| medium | 0.837 | 0.735 | 0.745 | 0.701 | 0.514 | 0.681 |
| large | 0.947 | 0.915 | 0.92 | 0.911 | 0.792 | 0.843 |
| xlarge | 0.961 | n/a | n/a | n/a | 0.935 | 0.95 |

masking). According to our explanations from Section 6, the use of fewer (text) tokens leads to a larger modality gap due to the greater information imbalance. Interestingly, despite this, downstream performance still improves on common benchmarks, as shown by Li et al. (2023a). When considering CLIP and CLIP-A models separately, we do find negative rank correlations: -0.667 (p=33.3%) for CLIP models and -0.6 (p=0.136) for CLIP-A models for RMG. When using L2M instead, we find 0.0 (p=1.0) for CLIP models and -0.6 (p=0.136) for CLIP-A models.

**Impact of data filtering.** There are two ways to reduce information imbalance from a data perspective: (1) caption enrichment and (2) filtering out high information imbalance image-text pairs, i.e., pairs with high uncertainty between them. We have shown extensive results for the former in Section 6 and Appendix F. To study the latter, we used CLIP models trained on CommonPool (Gadre et al., 2023), a benchmark to assess data filtering approaches.

Table 4 shows that data filtering is also effective in reducing the modality gap.

### D.3 IMAGENET RESULTS FOR FIGURE 4

Figure 11 shows similar results on ImageNet to the ones shown in the main text in Figure 4 on MS COCO.

Figure 12 shows that the modalities are also well separable when we select embedding dimensions based on the largest mean differences; similar to Figures 4b and 11c.

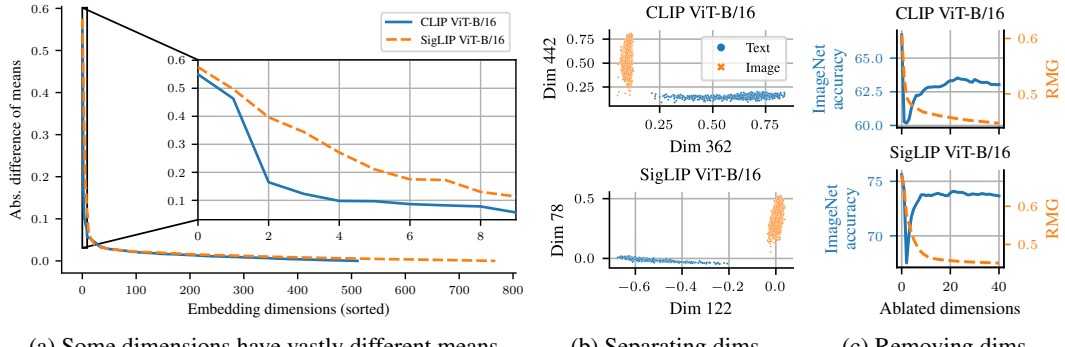

(a) Some dimensions have vastly different means.     (b) Separating dims.     (c) Removing dims.

Figure 11: **Few embedding dimensions separate the modalities.** Results on ImageNet. (a) We plot the absolute difference in the means of each embedding dimension between the modalities. Most dimensions have similar means for both modalities, but for some the differences are huge. (b) Pairs of these high difference dimensions can perfectly separate the modalities. (c) Successive removal of embedding dimensions based on the sorting of embedding dimensions from (a) leads to a sharp drop, followed by a partial recovery of downstream performance, while the modality gap gradually closes (similar results for L2M).

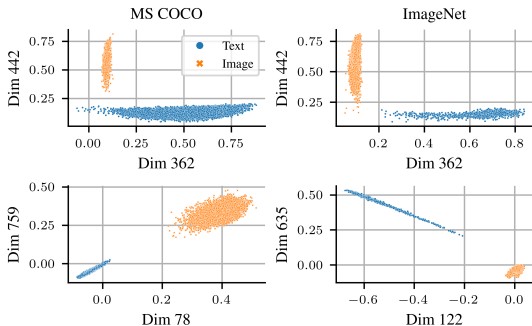

Figure 12: **Pairs based on the largest mean differences.** In the main paper, we show the dimensions with largest means. Note that using the largest mean difference also leads to separable dimensions. Top: OpenAI's CLIP ViT-B/16, bottom: SigLIP ViT-B/16. Left: MS COCO, right: ImageNet. In Figures 4b and 11c, we selected the embedding dimension of each modality, as each of the embedding dimensions most dominate the direction of the embedding vector of each modality.

### D.4 EXAMPLE FOR THE EFFECT OF THE EMBEDDING DIMENSION REMOVAL

Consider the following example: let $\mathbf{x} = [8, 0.6, 0.7, 0.3]^T$ be an image embedding, and $\mathbf{y} = [5, 0.13, 0.035, 0.02]^T$ and $\mathbf{y}' = [5, 1.5, 0.7, 0.45]^T$ be the matching or non-matching text embedding, respectively. We have cosine similarities $d$ of $d(\mathbf{x}, \mathbf{y}) = 0.995$ and $d(\mathbf{x}, \mathbf{y}') = 0.975$. After ablating the first dimension, we have cosine similarities of $0.822$ and $0.917$ and hence the image-text alignment flipped. Finally, it flips back after ablating also the second dimension.

### D.5 CLOSING THE GAP FOR IDEAL WORDS

We showed that we can close the modality gap post-hoc by successive removal of embedding dimensions in Section 4.2 but performance did not improve. Similarly, Liang et al. (2022) shifted the embeddings by the modality gap vector

$$\overrightarrow{\Delta}_{\text{gap}} := \frac{1}{N} \sum_{i=0}^{N} \mathbf{x}_i - \frac{1}{N} \sum_{i=0}^{N} \mathbf{y}_i \quad , \tag{5}$$

where $\mathbf{x}_i, \mathbf{y}_i$ are the $i$-th L2-normalized image or text embeddings, respectively. Simple shifting closed the modality but also did not improve performance.[4] But what's the effect on ideal words (Trager et al., 2023)?

**Ideal words.** Ideal words (Trager et al., 2023) can be computed as follows:

$$\mathbf{y}_{z_i} := \frac{1}{\alpha_{z_i}} \sum_{z'=(z'_1,\ldots,z'_k) \text{ with } z'_i=z_i} \beta_z \mathbf{y}_{z'} - \mathbf{y}_0 \quad , \tag{6}$$

where $z_i$ is a component of the factored set of concepts $z \in \mathcal{Z} = \mathcal{Z}_1 \times \ldots \times \mathcal{Z}_k$ each with $n_i$ possible concept values, $\mathbf{y}$ is the L2-normalized text embedding, $\alpha_{z_i} = \frac{1}{n_i}$, $\beta_z = \prod \frac{1}{n_i}$, and $\mathbf{y}_0$ is defined as

$$\mathbf{y}_0 := \beta_z \sum_z \mathbf{y}_z \quad . \tag{7}$$

In words, ideal words are computed by marginalizing out all other factors $j \neq i$. Intuitively, we can understand the ideal word $\mathbf{y}_{z_i}$ for component $z_i$ as the direction for that concept (e.g., "green") in the embedding space. Note that $y_0$ is simply the average text, which is used to remove the "textness" from the embedding vector.

**Extension to ideal images.** For our analysis, we propose to extend the notion of ideal words to ideal images by simply replacing the text embeddings, $\mathbf{y}$, by the image embeddings, $\mathbf{x}$, and utilize datasets with object-attribute information (MIT-States, UT-Zappos) to control $z$.

**Results.** Ideally, the ideal words and ideal images should align, i.e., the directions in the embedding space have a similar meaning. The cosine similarities between ideal words and ideal images in Table 5 show that ideal words and ideal images are roughly aligned. Once we correct for the modality gap, they are well aligned. However, Table 5 shows that the shifting has negligible to no effect on performance. This indicates that the removal of the main elements of the modality gap improves the cosine similarities (by removing the parts that lead to a low cosine similarity) but it does not fix the underlying problem, i.e., the modality gap does not limit performance of pre-trained contrastive VLMs.

# E EXTENDED DETAILS FOR SECTION 5

## E.1 FURTHER DETAILS ON OUR OBJECT ATTRIBUTE BIAS METRIC MOAD

For readability and to provide a better intuition about the metric we substituted parts with more informative variables in the main text. The definition of MOAD without substitution is given by:

$$\begin{aligned}
\text{MOAD}_{\text{img}} := \frac{1}{2|O|} \sum_{o \in O} \left( \frac{1}{N_1} \sum_{\substack{\mathbf{x}_i, \mathbf{x}_j \in X_o \\ i \neq j}} \mathbf{x}_i^T \mathbf{x}_j - \frac{1}{N_2} \sum_{\mathbf{x}_i \in X_o, \mathbf{x}_j \in X_{\neg o}} \mathbf{x}_i^T \mathbf{x}_j \right) \\
- \frac{1}{2|A|} \sum_{a \in A} \left( \frac{1}{N_3} \sum_{\substack{\mathbf{x}_i, \mathbf{x}_j \in X_a \\ i \neq j}} \mathbf{x}_i^T \mathbf{x}_j - \frac{1}{N_4} \sum_{\mathbf{x}_i \in X_a, \mathbf{x}_j \in X_{\neg a}} \mathbf{x}_i^T \mathbf{x}_j \right) \quad ,
\end{aligned} \tag{8}$$

where $x_i$ denotes the normalized image embeddings, $N_1, \ldots, N_4$ are normalization factors given by $N_1 = |X_o|^2 - |X_o|$, $N_2 = |X_o \times X_{\neg o}|$, $N_3 = |X_a|^2 - |X_a|$ and $N_4 = |X_a \times X_{\neg a}|$, $X_o, X_{\neg o}, X_a, X_{\neg a}$ are all image embeddings $\mathbf{x}$ that (not) entail the object $o \in O$ or attribute $a \in A$, respectively.

---

[4] Only when they optimized the shift length, they could improve performance.

Table 5: **Shifting ideal words makes them more similar to ideal images but does not improve performance.** We report the cosine similarities between ideal words and ideal images, and top-1 test accuracy on attributes and objects for CLIP ViT-B/16 (1st, 2nd table) and SigLIP ViT-B/16 (3rd, 4th) on MIT-States (1st, 3rd) and UT-Zappos (2nd, 4th). Note that we subtracted $\mathbf{y}_0$ for "ideal words" and for "ideal words with $\overrightarrow{\Delta}_{\mathrm{gap}}$" we did not. The reason for this is (approximate) equivalence between subtracting $\mathbf{y}_0$ and adding $\mathbf{x}_0$ to the modality gap vector $\overrightarrow{\Delta}_{\mathrm{gap}}$.

| | CLIP ViT B/16 on MIT-States | | |
|---|---|---|---|
| Variant | cosine similarity between | top-1 test accuracy | |
| | ideal words & ideal images | attribute | object |
| real words | n/a | 18.65 | 44.69 |
| ideal words | 0.388 | 19.74 | 46.87 |
| ideal words with $\overrightarrow{\Delta}_{\mathrm{gap}}$ | 0.858 | 19.70 | 46.84 |

| | CLIP ViT B/16 on UT-Zappos | | |
|---|---|---|---|
| real words | n/a | 15.92 | 60.54 |
| ideal words | 0.381 | 21.83 | 62.46 |
| ideal words with $\overrightarrow{\Delta}_{\mathrm{gap}}$ | 0.898 | 21.76 | 62.42 |

| | SigLIP ViT B/16 on MIT-States | | |
|---|---|---|---|
| real words | n/a | 21.82 | 50.23 |
| ideal words | 0.456 | 25.81 | 52.54 |
| ideal words with $\overrightarrow{\Delta}_{\mathrm{gap}}$ | 0.858 | 25.73 | 52.66 |

| | SigLIP ViT B/16 on UT-Zappos | | |
|---|---|---|---|
| real words | n/a | 27.14 | 73.27 |
| ideal words | 0.439 | 35.38 | 72.07 |
| ideal words with $\overrightarrow{\Delta}_{\mathrm{gap}}$ | 0.870 | 35.14 | 72.00 |

### E.2 GENERAL FORMULATION FOR BIAS MEASURE VIA RELATIVE ANGLE CHANGE OF EMBEDDINGS (BRACE)

MOAD is an instantiation of our novel metric Bias measure via Relative Angle Change of Embeddings (BRACE). BRACE is defined as follows:

$$
\mathrm{BRACE}_{\mathrm{img}} := \frac{1}{2|B|} \sum_{b \in B} \left( \frac{1}{N_1} \sum_{\substack{\mathbf{x}_i, \mathbf{x}_j \in X_b \\ i \neq j}} \mathbf{x}_i^T \mathbf{x}_j - \frac{1}{N_2} \sum_{\mathbf{x}_i \in X_b, \mathbf{x}_j \in X_{\neg b}} \mathbf{x}_i^T \mathbf{x}_j \right)
$$
$$
- \frac{1}{2|C|} \sum_{c \in C} \left( \frac{1}{N_3} \sum_{\substack{\mathbf{x}_i, \mathbf{x}_j \in X_c \\ i \neq j}} \mathbf{x}_i^T \mathbf{x}_j - \frac{1}{N_4} \sum_{\mathbf{x}_i \in X_c, \mathbf{x}_j \in X_{\neg c}} \mathbf{x}_i^T \mathbf{x}_j \right) , \tag{9}
$$

where $x_i$ denotes the normalized image embeddings, $N_1, ..., N_4$ are normalization factors given by $N_1 = |X_b|^2 - |X_b|$, $N_2 = |X_b \times X_{\neg b}|$, $N_3 = |X_c|^2 - |X_c|$ and $N_4 = |X_c \times X_{\neg c}|$, $X_b, X_{\neg b}, X_c, X_{\neg c}$ are all image embeddings $\mathbf{x}$ that (not) entail some group of concepts $B$ (e.g., objects $O$) or some other group of concepts $C$ (e.g., attributes $A$), respectively.

Table 6: **We define four conditions with various prior distributions over words $p(\text{word})$.** Throughout all conditions, digits are treated as the objects. In condition **A**, we ensure that each word appears the same number of times in the dataset, meaning $p(\text{word})$ is uniform. Note that $p(\text{word}_{\text{digit}}|\text{image}) = 1$ and $p(\text{word}_{\neg\text{digit}}|\text{image}) < 1$. In condition **D**, all factors appear in the caption, meaning $p(\text{word}|\text{image}) = 1$. Conditions **B** and **C** interpolate between conditions **A** and **D**. Note that the appearance probability (appearance counts in the dataset) of attributes $p(\text{word})$ is higher than for objects for conditions **B**, **C**, and **D**.

| | | | resulting $p(\text{word})$ for given $p(\text{word}|\text{image})$ | | | |
|---|---|---|---|---|---|---|
| | object | | attribute | | | |
| Condition | digit | thickthinning | swelling | fracture | scaling | color |
| **A** | 0.1 | 0.1 | 0.1 | 0.1 | 0.1 | 0.1 |
| **B** | 0.1 | 0.177 | 0.233 | 0.233 | 0.233 | 0.114 |
| **C** | 0.1 | 0.255 | 0.366 | 0.366 | 0.366 | 0.129 |
| **D** | 0.1 | 0.33 | 0.5 | 0.5 | 0.5 | 0.143 |

Table 7: **Even though attributes are globally more frequent (see Table 6), the object bias persists.** We observe object bias, even though $p(\text{word})$ is uniform (for **A**) or attributes are more frequently present in the captions (for **B**-**D**).

| | MOAD $_{\text{img}}$ | | |
|---|---|---|---|
| Condition | emb size 12 | emb size 15 | emb size 18 |
| **A** | 0.41 | 0.40 | 0.34 |
| **B** | 0.34 | 0.25 | 0.31 |
| **C** | 0.29 | 0.30 | 0.29 |
| **D** | 0.14 | 0.17 | 0.22 |

### E.3 ADDITIONAL EXPERIMENT ON WHERE THE OBJECT BIAS STEMS FROM

**Experimental setup.** Complementary to the experiment in Figure 5, we conducted experiments for which we controlled $p(\text{word})$ and $p(\text{word}|\text{image})$ on MAD. We considered four conditions **A**-**D** (see Table 6) that varied $p(\text{word})$ for objects and attributes.

**Results.** Table 7 shows that the models still exhibit a bias towards digits, even though attributes occur more frequently globally (in the dateset). This further confirms that the prior distribution over words $p(\text{word})$ cannot be the main factor for the emergence of the bias towards objects.

## F EXTENDED DETAILS FOR SECTION 6

### F.1 CC12M TRAINING AND INFORMATION IMBALANCE CAUSES THE MODALITY GAP EXPERIMENTS

**Experimental setup.** We largely followed the training setup of Ilharco et al. (2021). As common for CC12M, we used a ResNet-50 image encoder. We trained for 30 epochs, with each epoch having 9263104 image-text samples. We used a global batch size of 1760 and a learning rate of 1e-3 with 1000 steps of warm-up. For all other settings, we used the default values of Ilharco et al. (2021).

We trained CLIP models in 3 conditions:

- **full caption**: using the complete captions.
- **half caption**: we randomly dropped the first or second half of a caption.
- **quarter caption**: we only kept a randomly selected quarter of the caption.

Note, that we do not sample words randomly, but select a contiguous string of words from the caption. All hyperparameters are identical between the conditions.

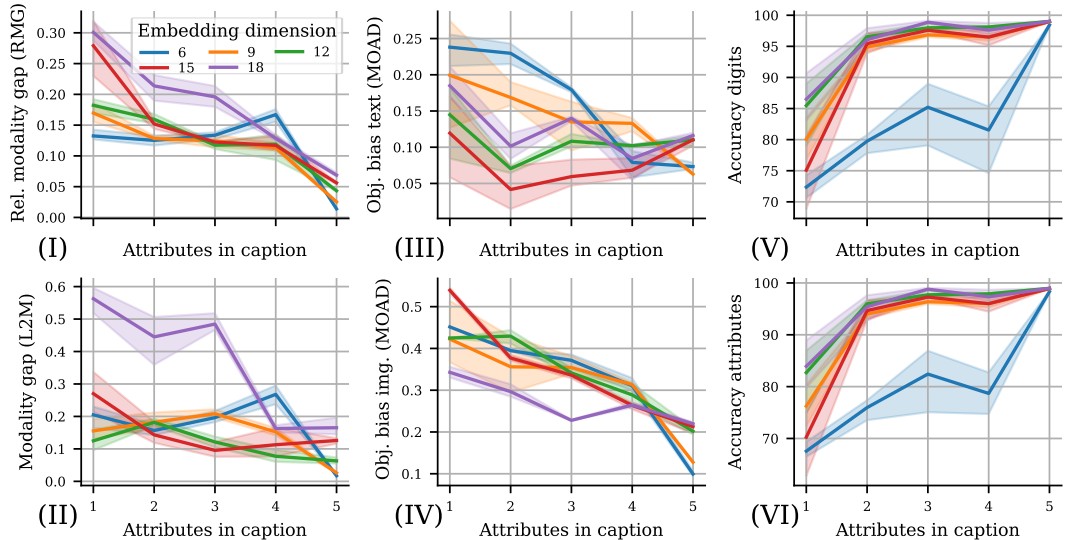

Figure 13: **Complete version of Figure 7a.** To study the influence of information imbalance between the modalities, we control the number of attributes present in the captions (the image is always affected by all attributes) in MAD. As the amount of information shared between the modalities increases, the modality gap (I-II) and bias towards objects reduces (III-IV), while downstream accuracy improves (V-VI).

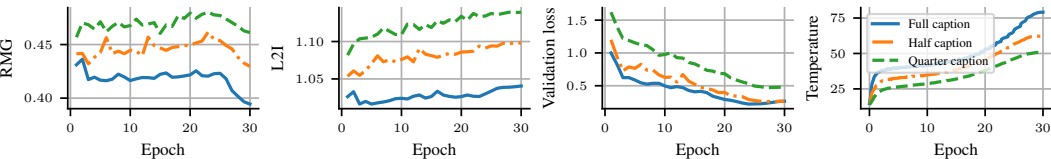

Figure 14: **Complete version of Figure 7c.** Training with "full" captions leads to a smaller modality gap (RMG, L2I), lower validation loss (though, "full" and "half" are very similar at the end of training), and higher temperature (i.e., more peaky distribution/less uncertainty).

**Additional results.** In addition to Figure 7c in the main paper, Figure 14 shows the L2-instance (L2I) modality gap metric and temperature. L2I could be an alternative to RMG, which is a compromise between RMG and L2M. In contrast to L2M, it computes the distance only where it matters, i.e., between the matching pairs. However, in comparison to RMG, it does not take into account the fraction of the embedding space used by the model. L2I is given by

$$L2I := \frac{1}{N} \sum_{i=1}^{N} ||\mathbf{x}_i - \mathbf{y}_i||, \tag{10}$$

where $\mathbf{x}_i$ is the $i$-th L2-normalized image embedding and $\mathbf{y}_i$ the $i$-th text embedding. Note that in contrast to RMG, L2I misses the drop of the relative gap (approx. epoch 25 in Figure 14), as the distance between the positives keeps increasing.

Figure 15 shows results for a different vision encoder (ViT-S in (a)), an alternative information-dropping scheme (sentence-dropping in (b)), and another dataset (CC3M in (c)). For all these variations, we find that the modality gap is smaller with less information imbalance ("full" caption) similar to our findings in Figures 7c and 14.

F.2 FURTHER EXPERIMENTAL DETAILS FOR SECTION 6.2

We used the pre-trained "full" caption model from Appendix F.1. Next, we fine-tuned this model using the same hyperparameters, but with "quarter" captions, i.e., keeping a contiguous quarter of

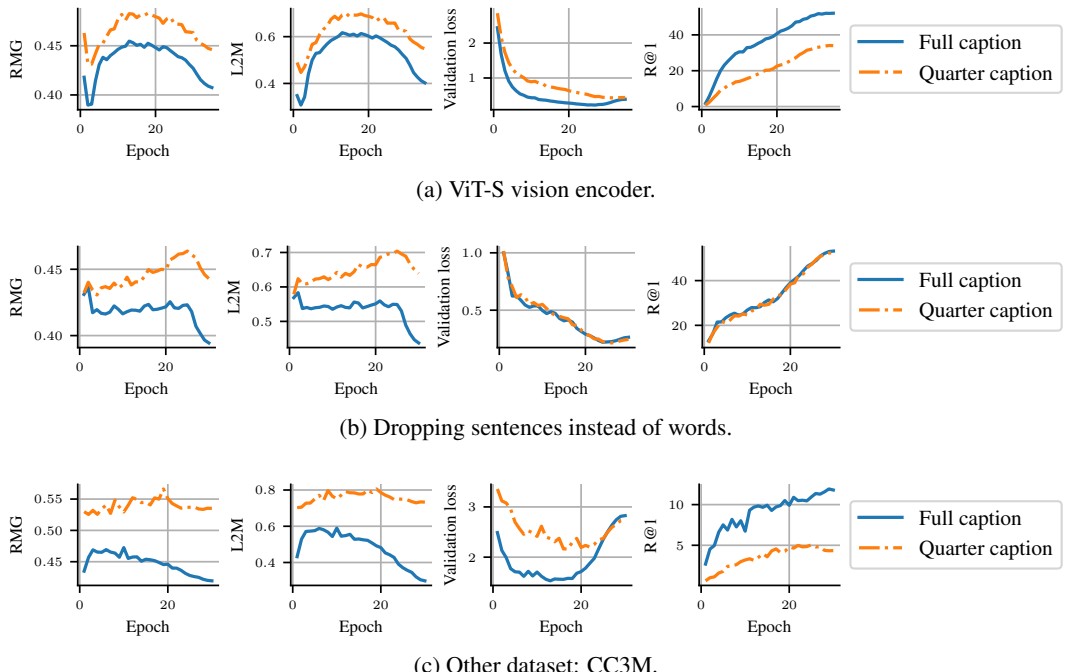

(a) ViT-S vision encoder.

(b) Dropping sentences instead of words.

(c) Other dataset: CC3M.

Figure 15: **Further evidence for the effect of information imbalance on the modality gap on real data.** This figure supports the findings of Figures 7c and 14 with experiments using (a) **a different vision encoder** backbone, (b) **sentence-wise information-dropping** scheme, and (c) **training on a different dataset** (note that CC3M is not a subset of CC12M). In all settings, "full" captions (i.e., less information imbalance) also leads to a smaller modality gap (RMG, L2M).

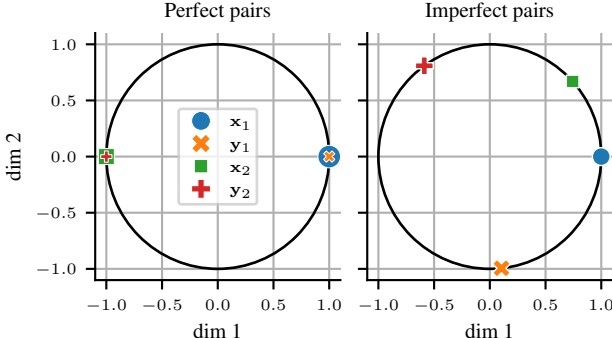

Figure 16: **Visualization of the global minima of the contrastive loss** under perfect or imperfect pairs in our toy 2D example. For perfect pairs (left), matching pairs are aligned and uniformly distributed. For imperfect pairs (right), the global minimum exhibits a modality gap. Here, $\mathbf{x}_i$ denotes $i$-th sample from modality 1, e.g., images, and $\mathbf{y}_i$ $i$-th sample from modality 2, e.g., text.

the words in the captions. For the "frozen temperature" model we freeze the temperature to the value learned in pre-training (i.e., $\sim$80).

### F.3 CAN THE MODALITY GAP EVEN APPEAR IN THE GLOBAL OPTIMUM?

We show below that the modality gap can even be present in the global optimum under certain data properties. Under perfect image-text pairs, it is straightforward to see that the global minimum of the CLIP loss does not exhibit a modality gap. I.e., all matching image-text pairs are perfectly aligned and the pairs are uniformly distributed on the unit hypersphere. But what happens when alignment is bounded, e.g., due to miscaptioning, focus on other factors, or missing information?

Table 8: **Effect of information imbalance in the "idealized" experimental setting of Fahim et al. (2024)**. It is apparent that information imbalance has a large impact on the modality gap.

| information imbalance | training time | L2M | RMG | I2I |
|:---:|:---:|:---:|:---:|:---:|
| | after initialization | 0.0 | 0.007 | 1.0 |
| ✗ | after 1200 epochs | 0.058 | 0.006 | 1.0 |
| ✓ | | 0.265 (+0.207) | 0.149 (+0.143) | 1.0 (±0.0) |

To illustrate the effect of such bounded alignment on the final image and text embeddings, we designed a 2D toy example. We generated two sets of points on the unit circle and directly optimized their positions. The points represent the embeddings of both modalities: $\{\mathbf{x}_1, \mathbf{x}_2\} = \mathbf{X}$ and $\{\mathbf{y}_1, \mathbf{y}_2\} = \mathbf{Y}$, where $\mathbf{x}_1, \mathbf{y}_1, \mathbf{x}_2, \mathbf{y}_2 \in \{[a, b]^T \mid a^2 + b^2 = 1; a, b \in [0, 1]\}$. Further, we specified which point of set $\mathbf{X}$ matches to which point in set $\mathbf{Y}$. For the perfect matching setting, we considered the following pairs:

$$\{(\mathbf{x}_1, \mathbf{y}_1), (\mathbf{x}_2, \mathbf{y}_2)\} \quad . \tag{11}$$

Trivially, the global minimum (up to rotations) is: $\mathbf{x}_1 = \mathbf{y}_1 = [1, 0]^T$, $\mathbf{x}_2 = \mathbf{y}_2 = [-1, 0]^T$; see Figure 16 for a visualization.

However, what happens when the alignment is bounded, e.g., due to mismatches? For image-text pairs, this can happen if a human annotator miscaptions an image. It could also stem from differing focuses among human annotators on distinct aspects of the image. We considered the following pairs:

$$\{(\mathbf{x}_1, \mathbf{y}_1), (\mathbf{x}_1, \mathbf{y}_1), (\mathbf{x}_2, \mathbf{y}_2), (\mathbf{x}_2, \mathbf{y}_2), \underbrace{(\mathbf{x}_1, \mathbf{y}_2), (\mathbf{x}_2, \mathbf{y}_1)}_{\text{mismatches}}\} \quad , \tag{12}$$

where we need the additional matching pairs $(\mathbf{x}_1, \mathbf{y}_1)$ and $(\mathbf{x}_2, \mathbf{y}_2)$ to avoid the degenerated global minimum $\mathbf{x}_1 = \mathbf{y}_1 = \mathbf{x}_2 = \mathbf{y}_2$.

To search for the globally optimal embeddings for the points in Equation 12, we ran a grid search with an angular resolution of $6°$. We found the following global minimum (again up to rotations): $\mathbf{x}_1 = [1, 0]^T$, $\mathbf{y}_1 = [\cos 276°, \sin 276°]^T$, $\mathbf{x}_2 = [\cos 42°, \sin 42°]^T$, $\mathbf{y}_2 = [\cos 126°, \sin 126°]^T$; see Figure 16 for a visualization. It is apparent that the global minimum exhibits a modality gap.

We recognize that identification and removal of the mismatches in Equation 12 may be straightforward in practice. However, our goal is to illustrate the impact of such in a *simplistic* setting to provide an intuition on the behavior of the contrastive multi-modal loss.

### F.4 IS THE GAP PRIMARILY CAUSED BY THE CONTRASTIVE LOSS INSTEAD OF THE INFORMATION IMBALANCE?

Recently, Fahim et al. (2024) proposed that the modality gap is inherent to the two-encoder contrastive loss. However, how does this relate to our explanation that the modality gap stems from an information imbalance in the data (Section 6)? To answer this, we replicated their "idealized experiment" (with minor differences that are highlighted below) and show that information imbalance has a substantially larger effect than the "contrastive gap".

**Experimental setup.** Following Fahim et al. (2024), we trained the CLIP model on *image-image pairs*. Note that this is in contrast to standard CLIP models that are trained on image-text pairs. However, it removes the differences that may stem from modality-specific properties. Further, since we can use the identical image, it also eliminates mismatches and, more importantly, information imbalance. To train the CLIP model, we replaced the text encoder with a copy of the image encoder. Fahim et al. (2024) initially closed the modality gap using a translation vector (i.e., similar to the post-hoc approach tested by Liang et al. (2022)). While this closes the gap in terms of L2, it does not close the gap in terms of RMG, as the cone size might still be different and representations might be rotated (or have completely different neighborhood relations). To address these issues, we closed the gap at initialization by simply initializing both image encoders with the *same* weights.

We considered two settings:

Table 9: **The contrastive loss almost closes the gap.** We directly optimize the embeddings.

| Step | Loss | RMG |
|---|---|---|
| 0 | 7.3789 | 0.4987 |
| 1000 | 0.0060 | 0.0061 |
| 2000 | 0.0014 | 0.0060 |

- **Without information imbalance**: we ran experiments as described above. This is equivalent (except for the different initialization) to the setting of Fahim et al. (2024).

- **With information imbalance**: we randomly resized and cropped (using the default parameters) the second image half of the time and the other half leave the image untouched (i.e., no information imbalance). Otherwise, we ran experiments as described above.

Note that evaluation was always conducted without information imbalance.

**Training details.** We followed the experimental setting of Fahim et al. (2024). We used 2048 randomly selected images from the MS COCO validation set (they used the training set). We also trained the CLIP models with a batch size of 64 for 1200 epochs. Both image encoders use the ViT-B/32 architecture with a 512-dimensional embedding dimensionality. We used a learning rate of 1e-4 with AdamW (with weight decay of 0.1, $\beta_1 = 0.9$, $\beta_2 = 0.98$, and $\epsilon = 1e-6$) and cosine annealing learning rate scheduling. The maximum temperature was set to 100.0. We repeated trainings for five random seeds.

**Results.** Table 8 shows that we can replicate the result of Fahim et al. (2024): even though the model is initialized without a modality gap and achieves perfect performance, there is a small gap after training. However, we also find that an information imbalance leads to substantially larger modality gaps measured with RMG (same results for L2M). This validates that information imbalance is the main driving factor for the modality gap.

## F.5 CAN THE CONTRASTIVE LOSS CLOSE THE MODALITY GAP?

To further understand whether the contrastive loss is capable to close the modality gap, we created a simplified example and stripped away all unnecessary factors, i.e., the encoder networks, similar to Appendix F.3. We randomly drew 1000 L2-normalized image embeddings and their matching text embeddings with 8 dimensions each. We directly optimized the embeddings using the CLIP loss. We trained for 2000 steps with a learning rate of 0.01 using Adam.

Table 9 shows that the contrastive loss can indeed narrow the gap and does not preserve it.

## F.6 CAN THE MODALITY GAP BE REDUCED IN PRE-TRAINED MODELS?

To show that the modality gap can also be reduced for pre-trained contrastive VLMs, we fine-tuned OpenAI's CLIP ViT-B/16 (Radford et al., 2021) and SigLIP ViT-B/16 (Zhai et al., 2023) on the Densely Captioned Images (DCI) dataset (Urbanek et al., 2024). DCI consists of 7805 dense and highly-aligned image-text pairs. Thus, DCI has reduced information imbalance and, consequently, the modality gap should reduce as we fine-tune.

**Training details.** We fine-tuned both pre-trained VLMs for ten epochs with global batch size of 256 (distributed across four NVIDIA RTX 2080 GPUs) with a warm-up of 100 steps, and a learning rate of 1e-5 with AdamW (with weight decay of 0.1, $\beta_1 = 0.9$, $\beta_2 = 0.09$, and $\epsilon = 1e-6$) and cosine annealing learning rate scheduling. The maximum temperature was kept at 100.0.

**Results.** Table 10 shows that training on the image-text pairs of DCI indeed reduces the modality gap, as expected by our arguments in Section 6.1. Note that fine-tuning led to slightly worse zero-shot transfer accuracy on ImageNet. This is not in conflict with our claims, as other factors such as the distributional shift between the fine-tuning and test datasets play a larger role for performance. For example, we do not expect better performance when fine-tuning on an image-text flower dataset and then evaluate on an image-text animal dataset. We suspect that DCI similarly may not be best suited for such object-centric benchmarks (ImageNet & MS COCO).

| | MS COCO | | | |
|---|---|---|---|---|
| | CLIP ViT-B/16 | | SigLIP ViT-B/16 | |
| | before fine-tuning | after fine-tuning | before fine-tuning | after fine-tuning |
| L2M | 0.816 | 0.700 | 1.046 | 0.958 |
| RMG | 0.572 | 0.499 | 0.623 | 0.562 |
| R@1 | 52.84 | 59.24 | 67.68 | 67.58 |
| | ImageNet | | | |
| L2M | 0.864 | 0.747 | 1.116 | 1.04 |
| RMG | 0.606 | 0.531 | 0.683 | 0.63 |
| Top-1 accuracy | 66.75 | 65.13 | 75.61 | 73.87 |

Table 10: **Fine-tuning on the image-text pairs of DCI** (Urbanek et al., 2024) **reduces the modality gap (lower L2M & RMG).**

## G   LIMITATIONS

We validated our main hypotheses on small- (MAD, CC3M) and medium-sized (CC12M) datasets. While additional experiments on large-scale datasets like LAION would be valuable, they are beyond our computational resources, unfortunately. Nevertheless, our analysis of pre-trained large-scale CLIP models aligns with our findings on smaller datasets, suggesting consistency across scales.

Additionally, our method of reducing the information imbalance on real data (keeping only a contiguous sequence of words) may not be optimal. For instance, dropping 50% of words should, on average, remove 50% of information, but the variance may be large. A more refined method could produce even clearer results.

Lastly, the results in Tables 1 and 3 are correlation-based. Due to the small number of comparable models, the findings in Table 3 lack statistical significance. Expanding our current analysis would require prohibitively large compute resources, which we unfortunately have not.

## H   CONNECTION TO PARTIAL MULTI-LABEL LEARNING

Information imbalance highlights the learning difficulties that arise from incomplete or partial information. In our case, a modality gap and object bias. This issue is also present in more classical multi-label learning. Specifically, the field of partial multi-label learning (Xie & Huang, 2018; Struski et al., 2023; Hang & Zhang, 2023) handles scenarios where an unknown number of ground truth labels are missing while irrelevant or adulterated ground truth labels are provided.

In this work, we showed that CLIP-like models face analogous challenges that stem from information imbalance. For example, information is missing from the captions, or captions entail mismatches. We believe that future work could study the similarities and potentially transfer ideas from partial multi-label learning to CLIP-like models, or vice versa.

