# OpenReview forum: "Two Effects, One Trigger: On the Modality Gap, Object Bias, and Information Imbalance in Contrastive Vision-Language Models"
_ICLR.cc/2025/Conference — ICLR 2025 Oral_

### Official Review · Reviewer_6t26 · 2024-10-25

**Soundness:** 3
**Presentation:** 2
**Contribution:** 3
**Rating:** 8
**Confidence:** 4

**Summary:**

The paper investigates two potential causes of the inferior performance of contrastive VLMs on downstream attribute recognition (compared to the often good downstream object recognition performance), namely, modality gap and object bias. The paper suggests that both modality gap and object bias are not directly related to the inferior performance problem, though they claim that with better controls modality gap seems to somewhat related.
Finally, the paper does controlled experiments on small-scale dataset to confirm that information imbalance (i.e., the concepts that are present in image but not in the corresponding captions) in a dataset is strongly correlated with modality gap and object bias. Decreasing information imbalance seems to give better downstream attribute recognition performance.

**Strengths:**

1. The paper provides in-depth investigation of the relationship between the inferior performance of contrastive VLMs on downstream attribute recognition and two potential causes (modality gap and object bias). They show that the relation is not substantial (though they claim that with better control the relation on modality gap exists, but I am not personally convinced given their evidence). This is informative for people studying these two factors. Importantly, they demonstrate that information imbalance might be a more important (causal) factor to investigate.
2. They give evidence for reducing the information imbalance leads to better attribute recognition performance, as a solution to this problem on contrastive VLMs. They also provide a way to do this by using trainable temperature.
3. The experiments presented in the paper are in general well-controlled.

**Weaknesses:**

1. To me, the major claim in the paper is that information imbalance causes both modality gap and object bias, and that even though these two are not substantially correlated with attribute recognition performance, decreasing information imbalance mitigates them and seemingly increase the overall attribute recognition performance. However, the paper spends most of the content to demonstrate details on modality gap and object bias, many of which are overly detailed. For example: in section 4.2, the authors talk about many attempts to naively close the modality gap, all of which fails to keep the model performance. I think contents like this can be briefly summarised and the details can be put into supplementary materials, since these details might diverge into information that doesn't help readers grasp the core of the main claim.
2. The major claim is interesting and informative. However, the evidence of it is mainly done on the simple synthesised dataset MAD. The authors give evidence on real-world datasets in Figure7 (c), but the evidence on RMG is much weaker (if significant at all) compared to that with MAD. Since this is evidence for the major claim, the author might consider adding more experiments (more model architectures, more datasets, and presumably a better way to control the number of attributes) for this.
3. The authors claim that “when we control for these factors, we observe the expected negative correlation, i.e., a smaller gap seems to correlate with better performance” (L245). However, the data they provided cannot support this claim, even when treating the significance values loosely. Some training datasets (e.g., DataComp-1B, OpenAI) clearly give positive correlation. The authors should consider removing this claim or providing more substantial evidence.
4. The overall presentation of the paper is not ideal, in that many “questions” (e.g., “Is object bias explained by the global word frequencies of the dataset?”) can be turned into statements. The question-answering patterns in the writing highlight the motivations. However, in this paper there are so many of them, and the text devoted for completing this pattern probably does not worth it. The author might consider using more statements instead.

**Questions:**

1. I think detailed contents on modality gap and object bias can be put into supplementary materials, since these details might diverge into information that doesn't help readers grasp the core of the main claim.
2. The author might consider adding more experiments (more model architectures, more datasets, and presumably a better way to control the number of attributes) to show that RMG also decreases with less information imbalance.
3. The authors should consider removing the claim “when we control for these factors, we observe the expected negative correlation, i.e., a smaller gap seems to correlate with better performance” or providing more substantial evidence.
4. The author might consider converting most of the question-answering patterns in the paper to statements.
5. The field of "Partial Multi-Label Learning" seems to be relevant to this paper (as another way to solve the information imbalance). I wonder if the authors are aware of this and if this will give a better solution than trainable temperature.

---

> ### Author Response · Authors · 2024-11-19
> **Response**
>
> We thank the reviewer for the thorough review and the constructive feedback. Below, we address the remaining concerns and questions.
>
> > (W1, Q1) Why does the paper “spend most of the content to demonstrate details on modality gap and object bias”?
>
> We believe there might be a misunderstanding. The primary goal of our work is to deepen our understanding of both the modality gap and object bias, see our title and TL;DR. Thus, the in-detail analysis is a fundamental part of our work.
>
> > (W2, Q2) The major claim is interesting and informative. However, the evidence of it is mainly done on the simple synthesised dataset MAD. The authors give evidence on real-world datasets in Figure7 (c), but the evidence on RMG is much weaker (if significant at all) compared to that with MAD. Since this is evidence for the major claim, the author might consider adding more experiments (more model architectures, more datasets, and presumably a better way to control the number of attributes) for this.
>
> Results on synthetic data naturally appear clearer than on real data since we have full control over the information imbalance. However, even in the real-word setting (where we only have partial control over this factor), Fig. 7c shows that the modality gap consistently decreases with less information imbalance (“full” captions) throughout the entire training.
>
> To further strengthen our claim, we have added experiments using (1) a different model architecture (ViT vision encoder in Fig. 14a), (2) a different image-text dataset (CC3M in Fig. 14c), and (3) an alternative way to control the number of attributes (sentence-dropping in Fig. 14b). These additional results strongly reinforce our conclusion that reducing information imbalance leads to a smaller modality gap.
>
>
> > (W3, Q3) Correlation statement in L245
>
> We do agree that the data provided does not support the claim that it correlates for all pre-training datasets. Explicitly, we use the phrase “seems to” correlate to express uncertainty. We further refer to the appendix, where we clearly state that these findings are not statistically significant and state that these results must be taken with a grain of salt, as they are not significant due to the small number of models. We acknowledge that the “seems to” in the main document can be easily missed and revised the sentence accordingly.
>
>
> > (W4, Q4) Overall presentation of the paper is not ideal in that many “questions” can be turned into statements.
>
> We believe that the question-answer pattern in the manuscript is a matter of stylistic preference. In our particular case, we even see it as advantageous, as in many cases turning the questions into statements would require substantial simplification, which can lead to misunderstandings. We already introduced the “take-aways” as statements that can be quickly read, but allow for more detail than section titles.
>
> However, if other reviewers also find this style problematic, we are open to rephrasing the text to a statement-driven format for improved clarity and presentation.
>
>
> > (Q5) Relation to the field of “partial multi-label learning”
>
> Thank you for pointing out this line of work. Partial multi-label learning indeed shares relevant challenges, particularly in addressing partial (label) information. We have added references to this line of work in Appendix H of the revised manuscript. We hope this will encourage further exploration of these ideas in the context of CLIP-like models and related frameworks. We believe that understanding and exploiting the similarities of our work and “partial multi-label learning” is a promising future direction.

---

> > ### Comment · Reviewer_6t26 · 2024-11-21
> > **Response to authors**
> >
> > My concerns are mostly addressed, which makes me lifting my previous rating.
> >
> > Regarding presentation, I see that the paper focused on analysis of this phenomenon. The analysis is comprised of multiple aspects and it is evaluable to show them altogether. From this perspective, I partially agree that the question-answer pattern is useful in that it is essentially hard to summarize the many detailed findings. But I still think many of the "takeaways" are compressible. I personally think that replacing the questions with the compressed version of these will make the article more concise.
> >
> > Regarding (W2, Q2), I have to say that previously I somehow misread the x-axis of figure 7 (c) to be "attributes in captions" rather than "epochs". Now with the correction and more data provided by the authors, the evidence from the real-data is more convincing. However, also as the authors point out, the attributes in real-world datasets is hard to control, so I will only consider the evidence to be slightly stronger.

---

> > > ### Author Response · Authors · 2024-11-22
> > > **Re: Response to authors**
> > >
> > > We thank the reviewer for their valuable feedback, which has helped us improve our work, as well as for the increased rating.

---

### Official Review · Reviewer_ZSLu · 2024-11-02

**Soundness:** 3
**Presentation:** 3
**Contribution:** 4
**Rating:** 8
**Confidence:** 3

**Summary:**

- This is an analysis paper, investigating the modality gap and object bias in VLMs. The authors argue that information imbalance is the root cause of both phenomena.
- A larger modality gap correlates superficially with better performance, but this is due to confounding factors like model size. Controlling for these shows that a smaller gap is beneficial.
- The authors introduce MOAD, a metric to measure object bias, and conduct experiments on synthetic and real-world datasets.
- They find that the modality gap is primarily driven by a few embedding dimensions and linked to entropy.
- Authors suggest that the model leverages the modality gap as a feature to manage prediction uncertainty.

**Strengths:**

- Clear and concise introduction of the problem, motivation, and takeaways.
- MOAD metric provides quantifiable measure of object bias.
- Well-designed experiments show compelling evidence for information imbalance hypothesis. Good coverage of 98 VLMs (e.g., CLIP and SigLIP).
- The proposed connection between the modality gap and entropy is intriguing

**Weaknesses:**

- There could be more analysis on real-world datasets to help solidify the claims.
- [Minor] There could be more in-depth discussion on the limitations of the findings.
- [Minor] There could be more explanation of mitigating the information imbalance.
- [Nitpick] Typo: "Extend details for Section 5" -> "Extended details for Section 5")

**Questions:**

Can you provide examples of caption enrichment strategies that could effectively reduce the information imbalance?

---

> ### Author Response · Authors · 2024-11-19
> **Response**
>
> We thank the reviewer for the thorough review and constructive feedback. Below, we address the remaining concerns and questions.
>
> > More analysis on real-world datasets to help solidify the claims
>
> To strengthen our findings from Fig. 7c (less information imbalance leads to smaller modality gap), we performed additional experiments. Specifically, we experimented with a different vision encoder backbone (ViT vision encoder in Fig. 15a), a different information-dropping scheme (sentence-dropping in Fig. 15b), and different dataset (CC3M in Fig. 15c). Combined with the real-world dataset results already in the manuscript, these experiments further solidify our claims.
>
> If the reviewer has more experiments in mind to further solidify our findings, we are happy to incorporate them.
>
> > More in-depth discussion on the limitations of the findings
>
> We validated our main hypotheses on small- (MAD, CC3M) and medium-sized (CC12M) datasets. While additional experiments on large-scale datasets like LAION would be valuable, they are beyond our computational resources, unfortunately. Nevertheless, our analysis of pre-trained large-scale CLIP models aligns with our findings on smaller datasets, suggesting consistency across scales.
>
> Additionally, our method of reducing the information imbalance on real data (keeping only a contiguous sequence of words) may not be optimal. For instance, dropping 50% of words should, on average, remove 50% of information, but the variance may be large. A more refined method could produce even clearer results.
>
> Lastly, the results in Tabs. 1 & 3 are correlation-based. Due to the small number of comparable models, the findings in Tab. 3 lack statistical significance. Expanding our current analysis would require prohibitively large compute resources, which we unfortunately don't have.
>
> We acknowledge these limitations in Appendix G of the revised manuscript.
>
> > More explanation of mitigating the information imbalance; Can you provide examples of caption enrichment strategies that could effectively reduce the information imbalance?
>
> Human-enriched captions, as in the DCI dataset [1], are the most effective way to reduce information imbalance. Appendix F.6 shows that fine-tuning on such high-quality image-text pairs significantly reduces the modality gap. However, scaling human enrichment to millions of image-text pairs is a challenge.
>
> Therefore, it is crucial to explore automated approaches to allow for scale. A key requirement for any caption-enhancing approach is that it must be image-dependent. For example, LaCLIP [2] rewrites captions without looking at the images, which can and does lead to hallucinations. Thus, we have to leverage *multi-modal* LLMs that enrich the caption conditioned on the image. However, they come with known issues of self-labeling. Despite them, multi-modal LLMs represent one of the most scalable and practical approaches to reduce information imbalance.
>
> We see developing effective caption enrichment strategies as an intriguing area for future research. Our findings provide insights into the challenges of mitigating information imbalance and may serve as a foundation for developing and improving automated caption enrichment strategies.
>
> > Typo (“Extend details for Section 5”)
>
> Thanks, we fixed this in the revised manuscript.
>
> ---
>
> [1] Urbanek, Jack, et al. "A picture is worth more than 77 text tokens: Evaluating clip-style models on dense captions." CVPR 2024.
>
> [2] Fan, Lijie, et al. "Improving clip training with language rewrites." NeurIPS 2023.

---

> > ### Comment · Reviewer_ZSLu · 2024-11-23
> >
> > Thank you for your response. I have no further questions.

---

> > > ### Author Response · Authors · 2024-11-28
> > >
> > > We would like to thank you again for reviewing our work and for your feedback.

---

### Official Review · Reviewer_qe7i · 2024-11-04

**Soundness:** 4
**Presentation:** 3
**Contribution:** 3
**Rating:** 8
**Confidence:** 4

**Summary:**

This paper extensively investigates the modality gap.
Different from the previous works, which hypothesize the modality gap is caused by the cone effect at model initialization or object biases, the authors analyze the modality gap from a modality information imbalance perspective.
They hypothesize the information imbalance between the two modality data (i.e., image and text) causes the modality gap and object bias, and demonstrate the hypothesis through extensive empirical studies.
In addition, they provide an analysis that how models lead to the modality gap with imbalanced image-text data.
Summarized 7 takeaways and corresponding analysis provide a novel insight into the modality gap research.

**Strengths:**

**[S1]** This paper introduced a new perspective on the modality gap, providing rich insight to the researchers and practitioners.

**[S2]** The paper is well-written and extensive experimental analysis is sufficient to validate their hypothesis and argument.

**[S3]** Proposed metrics to measure the modality gap (i.e., RMG) and object/attribute biases (i.e., MOAD) are well designed to intuitively compare the corresponding components.

**[S4]** The provided appendix is ​​sufficient to follow the details and experimental setting of the paper, enabling a better understanding.

**Weaknesses:**

**[W1] Mismatch in the level of information imbalance between synthetic data and real data settings**
- The authors provide experimental validations on fully-controlled synthetic data and real data.
With synthetic data, information imbalance is defined based on the number of attributes, making it reasonable and sufficient to validate their hypothesis.
However, half/quarter captions of real data are derived by randomly dropping the part of captions.
I wonder if this strategy can hold the same hypothesis as in synthetic data. Please see Q2 below.

**[W2] The lack of evidence on actual pretrained VLMs**
- This paper suggests that the modality gap or the object bias can be reduced by data filtering or caption enrichment, and the performance can also be improved. However, there is a lack of analysis on real pretrained VLMs (e.g. pretrained CLIP and SigLIP) with large modality gaps. See Q3 below.

**Questions:**

**[Q1] L261**
- I think "with the largest mean" in L261 should be "with the largest mean differences."

**[Q2] Caption setup in Section F.1**
- The authors made the half and quarter captions by randomly dropping the part of a caption.
This approach may cause incompleteness in the caption itself rather than information imbalance.
Is there a strategy to verify the quality of the generated captions?
I think the paper would be better if there is an analysis using metrics that could verify that half/quarter captions are grammatically perfect but only contain different amounts of information.

**[Q3] Reverse process in Section 6.1**
- The authors provide the experimental validation by training the model with "full" captions and fine-tuning with "half" and "quarter" captions. It is reasonable and makes sense. But can we expect the same effect on the reverse process?
In other words, can the performance of the pretrained VLMs (e.g. CLIP, SigLIP in this paper), which have a large modality gap, be improved, while reducing the modality gap by fine-tuning on an enriched dataset?

---

> ### Author Response · Authors · 2024-11-19
> **Response**
>
> We thank the reviewer for taking the time to review our paper and the constructive feedback. Below, we address the remaining concerns and questions.
>
> > (W1 & Q2) The authors made the half and quarter captions by randomly dropping the part of a caption. This approach may cause incompleteness [or grammatical errors] in the caption itself rather than information imbalance.
>
> Our synthetic setting ensures precise control of the data without grammatical errors, allowing for a clean study of the effects of information imbalance. However, in real data, achieving the same level of control is challenging. To mimic the information imbalance settings on real-world datasets, we only keep a *contiguous* sequence of words in the captions (importantly, we do not drop random words). While this effectively controls the number of words, it can sometimes produce incomplete and grammatically incorrect captions (e.g., partial sentences). However, prior work [1, 2] indicates that CLIP models primarily learn Bag-of-Words representations and, consequently, are not heavily reliant on the finer grammatical structures. Thus, we believe this is less problematic than one may think.
>
> To confirm that our results indeed hold with “grammatically perfect” captions, we conducted an experiment where we dropped entire sentences. Fig. 15b in the revised manuscript shows that the modality gap also increases with increasing information imbalance (due to dropped sentences).
>
> >  (W2 & Q3) Can the performance of the pretrained VLMs (e.g. CLIP, SigLIP in this paper), which have a large modality gap, be improved, while reducing the modality gap by fine-tuning on an enriched dataset?
>
> We fine-tuned OpenAI’s CLIP ViT-B/16 and SigLIP ViT-B/16 on the high-quality image-text pairs of DCI [3]. Please refer to Appendix F.6 for the fine-tuning details and results.
> We find that fine-tuning on high-quality image-text pairs indeed reduces the modality gap and can lead to improvements. Note that we observe slightly lower performance for ImageNet, which is likely due to a distributional shift between DCI and ImageNet.
>
> > Q1: I think "with the largest mean" in L261 should be "with the largest mean differences."
>
> Thank you for bringing this to our attention. Note that what we wrote is correct. Showing the dimensions with the largest difference in means, as suggested, is another valid choice. For completeness, we added the plots using the dimensions with the largest difference in means in Fig. 12 of the revised manuscript. As expected, these dimensions also exhibit a clear separation between modalities. We revised the caption of Fig. 4 to make this more clear.
>
> ---
>
> [1] Tang, Yingtian, et al. "When are lemons purple? the concept association bias of vision-language models." EMNLP 2023.
>
> [2] Yuksekgonul, Mert, et al. "When and why vision-language models behave like bags-of-words, and what to do about it?." ICLR 2023.
>
> [3] Urbanek, Jack, et al. "A picture is worth more than 77 text tokens: Evaluating clip-style models on dense captions." CVPR 2024.

---

> > ### Comment · Reviewer_qe7i · 2024-11-24
> >
> > Thank you for your good work and responses.

---

> > > ### Author Response · Authors · 2024-11-28
> > >
> > > We would like to thank you again for reviewing our work and suggesting additional experiments that helped us improve it.

---

### Official Review · Reviewer_MYFm · 2024-11-07

**Soundness:** 3
**Presentation:** 3
**Contribution:** 3
**Rating:** 8
**Confidence:** 3

**Summary:**

This paper presents a comprehensive study of contrastive VLMs, particularly examining why they excel at object recognition but struggle with attribute detection. The paper investigates how the modality gap and object bias affect downstream performance, and why these phenomena emerge.
The study supports these conclusions through extensive experiments using various models (particularly CLIP ViT-B/16 and SigLIP ViT-B/16) and datasets.

**Strengths:**

I've reviewed this paper and noticed many improvements from the previous version:
- Takeaway 1 is now much clearer. The section effectively demonstrates how common confounding factors affect results, and shows that when they control for these factors, a smaller modality gap indeed leads to better performance in downstream tasks.
- The authors have introduced a new metric called RMG, which successfully addresses the limitations of L2M.
- Takeaway 2 is interesting, and it leads nicely into Takeaway 3, which shows that we can actually reduce the modality gap using post-hoc methods.
- Overall, this is a well-written study that thoroughly examines both the modality gap and object bias in contrastive vision-language models.

**Weaknesses:**

- I think most of weaknesses that have been seen in the previous submission was addressed.

**Questions:**

- For Takeaway 3, can you elaborate what "the modalities have different local neighborhoods" mean?

---

> ### Author Response · Authors · 2024-11-19
> **Response**
>
> We thank the reviewer for reviewing our paper again. We are pleased that the reviewer “noticed many improvements”. Below, we address your remaining question:
>
> > “For Takeaway 3, can you elaborate what "the modalities have different local neighborhoods" mean?”
>
> By different local neighborhoods, we mean that the neighborhood relations (e.g., which visual or text embeddings of one class are closest to another) are different between the modalities. For example, while the visual embeddings of class A may be closer to class B than class C, the text embeddings of class A might instead be closer to class C than class B.
>
> We hope this clarifies the reviewer’s question. If the reviewer has further concerns or additional questions, we would be happy to address them during the discussion phase.

---

> > ### Comment · Reviewer_MYFm · 2024-11-26
> >
> > Thank you for your response -- it clears up my question. I have no further questions.

---

> > > ### Author Response · Authors · 2024-11-28
> > >
> > > We are happy to hear that all your questions are cleared up. Thank you for reviewing our work and increasing the score.

---

### Author Response · Authors · 2024-11-19
**Global response**

We thank all reviewers for their time spent on reviewing our manuscript and for their constructive feedback.

Overall, the reviewers’ feedback is positive. Reviewers highlight that our paper offers “a new perspective on the modality gap, providing rich insight to the researchers and practitioners” (qe7i) and find our “proposed connection between the modality gap and entropy [...] intriguing” (ZSLu). They also find our study “well-written [...] that thoroughly examines both the modality gap and object bias in contrastive vision-language models” (MYFm), and the “experiments [...] are in general well-controlled” (6t26).

Below is a summary of the revisions of the manuscript and we highlight changes with blue text in the revised manuscript:

* Additional experiments on real-world datasets, varying model architecture, information-dropping scheme, and dataset, to further support our finding that less information imbalance leads to a smaller modality gap (Fig. 15).
* Fine-tuning experiments to reduce the modality gap of pre-trained models (Appendix F.6).
* Added scatter plots for Fig. 4b when choosing the dimensions based on the largest mean difference (Fig. 12)
* Added limitation section (Appendix G).
* Added connection to partial multi-label learning (Appendix H).
* Minor revisions, including clarifications and fixed typos based on reviewers’ suggestions.

---

### Meta-Review · Area_Chair_sFbH · 2024-12-19

**Metareview:**

This paper studies the phenomena of modality gap and object bias in contrastive VLMs, and shows that they stem from an information imbalance between modalities, limiting alignment in the embedding space, with the modality gap driven by few dimensions, linked to higher logit entropy, and object bias tied to caption presence bias.  It initially received scores of 5,6,8,8.  Positive points include intriguing study on the connection between the modality gap and entropy, a new perspective on the modality gap, thorough experimental analysis, and clear writing.  Negative points include needing more analysis on real-world datasets, lack of evidence on pretrained VLMs, some unsubstantiated claims, and more discussions on limitations.  The rebuttal and discussion adequately addressed these concerns.  Two reviewers increased their scores, so that the final score is 8,8,8,8.  After carefully considering the paper, reviews, rebuttal, and discussion, the AC feels that this paper is interesting and will provide a valuable contribution to the ICLR community, and recommends spotlight acceptance.

**Additional Comments On Reviewer Discussion:**

Positive points include intriguing study on the connection between the modality gap and entropy, a new perspective on the modality gap, thorough experimental analysis, and clear writing.  Negative points include needing more analysis on real-world datasets, lack of evidence on pretrained VLMs, some unsubstantiated claims, and more discussions on limitations.  The rebuttal and discussion adequately addressed these concerns.  Two reviewers increased their scores, so that the final score is 8,8,8,8.  After carefully considering the paper, reviews, rebuttal, and discussion, the AC feels that this paper is interesting and will provide a valuable contribution to the ICLR community, and recommends spotlight acceptance.

---

### Decision · Program_Chairs · 2025-01-22

Accept (Oral)